# TABPFN-WIDE: CONTINUED PRE-TRAINING FOR EXTREME FEATURE COUNTS

## ABSTRACT

Revealing novel insights from the relationship between molecular measurements and pathology remains a very impactful application of machine learning in biomedicine. Data in this domain typically contain only a few observations but thousands of potentially noisy features, posing challenges for conventional machine learning approaches. While prior-data fitted networks emerge as foundation models for tabular data, they are currently not suited to handle large feature counts $(> 500)$. Although feature reduction enables their application, it hinders feature importance analysis. We propose a strategy that extends existing models through continued pre-training on synthetic data sampled from a customized prior. The resulting model, TabPFN-Wide[1], matches or exceeds its base model's performance while exhibiting improved robustness to noise. It seamlessly scales beyond 50,000 features, regardless of noise levels, while maintaining inherent interpretability, which is critical for biomedical applications. Our results show that prior-informed adaptation is suitable to enhance the capability of foundation models for high-dimensional data. On real-world biomedical datasets many of the most relevant features identified by the model overlap with previous biological findings, while others propose potential starting points for future studies.

## 1 INTRODUCTION

Data stored in a table are an important data modality used for quantitative research in healthcare, finance, natural sciences, and many more. Tabular data are relevant for many real-world applications and "offer[s] uniquely exciting, large, unsolved challenges for researchers" (van Breugel & van der Schaar, 2024). One such challenge is high-dimensional, low-sample-size (HDLSS) data, for example, found in biomedical research. Cohort sizes of studies are small due to cost, time, or disease rarity, while modern biomedical technologies, on the other hand, enable the measurement of thousands of features per patient. Collected data can then be examined, for example, to study interactions between thousands of biomarkers and cancer types (McLendon et al., 2008; Bell et al., 2011). To guide scientific discovery (Ditz et al., 2023a;b), interpretability is as important as accuracy. Overall, such extreme feature counts in combination with a low sample size pose a challenge for real-world machine learning applications.

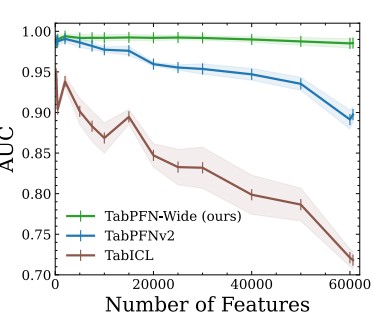

Figure 1: The performance of existing tabular foundation models decreases for a selected high-dimensional biomedical dataset. Further datasets are presented in Section 5.1 to confirm generality.

Foundation models for structured data have emerged, and models like TabPFN and TabICL (Hollmann et al., 2023b; 2025; Qu et al., 2025) are currently at the forefront of predictive tabular ML benchmark tasks (Erickson et al., 2025). These state-of-the-art models use in-context learning (ICL) (Brown et al., 2020) and are based on transformers, pre-trained on synthetic or real-world data to solve regression and classification tasks. As a result, they are highly effective on unseen

---

[1]Training code and model weights will be released upon acceptance.

tasks with characteristics similar to those seen during pre-training. While the exact training data are often unknown, empirical performance on HDLSS data (see brown and blue lines in the example in Figure 1) suggests that current models have not learned to handle extreme feature counts.

Such limits stem from insufficient exposure during pre-training and not necessarily from a lack of model capacity, data or resources; thus, re-training from scratch could be a solution. However, re-training from scratch whenever we encounter a new task or data characteristic to "fix" a model would be extremely resource-intensive, and therefore often infeasible. This also contradicts the concept of a "foundation model", pre-trained to serve as the basis for downstream tasks. Naive solutions, such as subsampling or compressing features to match the dimensionality of the pre-training data, render methods for quantifying feature importance ineffective. Instead, we aim to enhance the capability of existing models as a resource-efficient solution, while keeping the interpretability workflow intact. Concretely, we study the more general question: "How can continued pre-training extend tabular foundation models to generalize across diverse task types in high-dimensional, small-sample data?"

Specifically, our contributions are:

1. We develop a novel prior to efficiently generate synthetic HDLSS data.

2. We propose continued pre-training to extend TabPFNv2, resulting in TabPFN-Wide, to handle extreme feature counts beyond 50,000 features.

3. In empirical evaluations on biomedical data and standard tabular benchmark tasks, we show that TabPFN-Wide maintains performance within its original range, while being significantly more robust on wide data.

4. Finally, we study the inherent interpretability of TabPFN-Wide and show that attention maps allow us to identify relevant features.

## 2   PROBLEM DESCRIPTION

We start by briefly describing our problem setup and the challenges for robustly scaling tabular foundation models, specifically TabPFNv2 (Hollmann et al., 2025), to thousands of features.

**Tabular data** can be described as a dataset $\mathcal{D} = \{(\boldsymbol{x}_i, y_i)\}_{i=1}^{n}$ containing $n$ samples (rows). Each sample consists of a feature vector $\boldsymbol{x}_i \in \mathbb{R}^m$ with $m$ features (columns) and, for classification tasks, a corresponding label $y_i \in \{1, 2, ..., C\}$. To measure performance of a model $f$, we split available data into a train dataset $\mathcal{D}_{train} = \{(\boldsymbol{x}_i^{(train)}, y_i^{(train)})\}_{i=1}^{n_{train}}$ and a validation dataset $\mathcal{D}_{val} = \mathcal{D} \setminus \mathcal{D}_{train}$ and compute a loss, e.g., log loss, $\mathcal{L} = \sum_{(\boldsymbol{x}_i, y_i) \in \mathcal{D}_{val}} l\left(f(\boldsymbol{x}_i, \mathcal{D}_{train}), y_i\right)$ to approximate how well $f$ would generalize to unseen (test) samples. What distinguishes tabular data from other modalities are their heterogeneous feature types (categorical, numerical, missing values, etc.), and potentially diverse structures with the number of samples and features ranging from a few to millions (van Breugel & van der Schaar, 2024).

**HDLSS data** are a specific type of tabular data where the number of features is much larger than the number of samples, i.e., $m \gg n$. Such data typically occur in the biomedical domain. For example, cancer data from The Cancer Genome Atlas (TCGA) provide high-dimensional multi-omics measurements from cancer patients, such as those with ovarian cancer (Bell et al., 2011). In this setting, a typical classification problem is the identification of cancer subtypes. Improving the accuracy and robustness of predictive machine learning models supports precise diagnoses and personalized treatments, ultimately improving patient outcomes. A key difficulty arises from the high-dimensional feature space of molecular data, where noisy or irrelevant measurements often obscure subtype-specific signals. This complexity inhibits the detection of biologically meaningful patterns and hinders the ability to distinguish molecular differences between tumor subtypes.

**Interpretability** is crucial, especially for biomedical downstream tasks. However, for HDLSS data common post-hoc interpretability methods are unreliable (Bordt et al., 2022). For example, traditional permutation-based testing approaches like SHAP (Lundberg & Lee, 2017) require computing scores for each variable multiple times across multiple permutations, making it computationally infeasible for high-dimensional datasets. Additionally, the low sample size reduces the stability of the results.

Consequently, feature reduction or selection techniques are applied beforehand to reduce the number of features to a computable range. Yet, this inherently poses the risk of losing information or dropping potentially relevant features, which would be highly undesirable for applications in the real world. Thus, we avoid feature reduction and instead make our model work on all available features. This allows the model to identify the most predictive features directly. To gain insights into this internal selection process, we sought inherent interpretability methods and chose to use attention maps computed within the transformer architecture. However, the role and interpretability of attention maps are controversial in the literature, with nearly no previous work on attention analysis of TabPFN (or related models). In the context of large language models (LLMs), studies have shown that while attention maps may provide a coarse indication of a model's reasoning process, they are often noisy and can erroneously emphasize irrelevant tokens (Serrano & Smith, 2019; Jain & Wallace, 2019). Nevertheless, there have been interesting approaches in biomedicine, where features found by attention maps were supported by biological knowledge (Ditz et al., 2023a;b).

For TabPFNv2's attention specifically, earlier research shows that it evolves across layers, shifting from label-focused attention in the first layers to semantically relevant attribute attention in deeper layers (Ye et al., 2025). Additionally, Rubachev et al. (2025) link a reduced entropy of the attention score distribution to a more focused classification model. Building on these observations, we examine the attention maps as described, with careful consideration of their potential shortcomings.

## 3 Tabular Foundation Models for Predictive ML Tasks

**Prevailing models changed from traditional to pre-trained models.** Traditional ML models, like random forests or multi-layer perceptrons, must be trained from scratch for each task, with their predictive quality depending on hyperparameters and encoded inductive biases. With the rise of transformer models, amortized inference as a new learning paradigm for tabular data has emerged. Such models are trained across many (synthetic) datasets to *learn how to do statistical inference* via ICL. At inference time, training samples and query points are fed to the model, which then approximates Bayesian inference to predict labels (Müller et al., 2022; Müller et al., 2025).

The use of ICL for predictive tabular tasks was originally based on LLMs. Further building on the successes of LLMs, numerous studies have investigated their application to tabular data (Hegselmann et al., 2023; Zhang et al., 2024; Herzig et al., 2020). For these approaches, natural language representations of the tables are used for few- and zero-shot tabular classification. However, table-to-text-based models are limited by the context window of the underlying LLM; their predictions could be based on learned world knowledge rather than the table data, and, importantly, they cannot inherently leverage the structure (columns and rows) of tabular data. While yielding impressive results for zero- and few-shot tasks, they perform worse, when more data are available (Hegselmann et al., 2023). To address these weaknesses while simultaneously keeping the ICL approach, tabular foundation models emerged, with TabPFN (Hollmann et al., 2023a) being one of the earliest representatives. It is entirely trained on synthetic data generated from a prior based on structural causal models, yielding competitive performance on unseen tabular classification tasks. TabPFNv2 (Hollmann et al., 2025), a follow-up, introduced a modified prior and architecture, achieving state-of-the-art performance on datasets with up to 10,000 samples and 500 features.

**Current research focuses on extending the applicability regarding the number of samples and computational cost.** One prominent example is TabICL (Qu et al., 2025), which uses only a fixed number of embedded [CLS] tokens per sample for ICL rather than all the features. Furthermore, TuneTables (Feuer et al., 2024) optimizes the context of TabPFN using a learned compact dataset representation instead of the whole training data. Additionally, TabFlex (Zeng et al., 2025) uses linear attention instead of standard (quadratic) attention to reduce complexity. Other research directions focus on localization approaches to select relevant context samples (Ma et al., 2025; Xu et al., 2024; Koshil et al., 2024). While all these approaches aim to extend the application range, they propose new architectures and inference mechanisms, often applying feature reduction and compression. In contrast, we aim to expand an *existing* model's capability without impairing interpretability on a per-feature level. For these reasons, we focus on TabPFNv2 (Hollmann et al., 2025), currently the only state-of-the-art approach that can simply be modified (see Section 4.3) to satisfy our requirement of preserving a per-feature resolution throughout its architecture.

**Fine-tuning and continued pre-training improve performance on downstream tasks.** Fine-tuning, i.e., performing gradient updates using data from the target downstream tasks, is commonly used to adapt LLMs to application domains (Christophe et al., 2024; Weyssow et al., 2024) and has been proposed as a best practice to compare models (Zhang et al., 2025a). Similarly, fine-tuning TabPFN in general (den Breejen et al., 2025; Rubachev et al., 2025) or specifically performing parameter-efficient fine-tuning for context optimization (Feuer et al., 2024) can improve performance on a single downstream task. However, this requires a sufficient number of samples for this task. Continued pre-training, in contrast, does not use data from the target task but leverages tasks with properties similar to the target task. For example, Real-TabPFN (Garg et al., 2025), further pre-trained on real-world datasets, shows significant improvements on real-world tabular benchmarks. We follow this direction, but instead of using real-world data, we study how to continue pre-training with synthetic data to scale TabPFN to extreme feature counts, far beyond what it has seen during pre-training. Because this involves sequential training, it is crucial to prevent the model from experiencing catastrophic forgetting (French, 1993; Kemker et al., 2018). This could cause the model to perform significantly worse on tabular data within the original ranges of TabPFNv2.

## 4 METHODOLOGY

We propose a novel approach to extend the capability of tabular foundation models, like TabPFNv2, while preserving per-feature interpretability. We split our method into three components: First, we develop a prior to efficiently generate synthetic HDLSS data. Second, we use this data to continue pre-training and, third, we study attention maps for feature-wise interpretability.

### 4.1 A PRIOR FOR SYNTHETIC HDLSS DATA GENERATION

To adapt our model, we need a mechanism to generate training data, which (1) works fast and cost-effectively, since we need multiple datasets per batch step, and (2) yields realistic data, to provide a meaningful and reliable signal during adaptation.

**HDLSS prior.** For the first desideratum, we follow prior work and rely on synthetic data obtained from a data-generating mechanism based on structural causal models (Hollmann et al., 2023a;b). Datasets are therefore drawn from randomly sampled directed acyclic graphs. Specifically, as the TabPFNv2 prior is not publicly available, we use the open-source prior used to train TabICL (Qu et al., 2025), considering TabICL's strong empirical performance as evidence of the prior's similar effectiveness. To fulfill the second desideratum, we leverage the assumption that features in HDLSS data often exhibit substantial noise and strong inter-feature correlations (Clarke et al., 2008). Based on this, we construct a suitable prior as illustrated in Figure 2 (right).

---

**Algorithm 1** Feature Widening

**Input:** Input features $X \in \mathbb{R}^{n \times m}$ , target dimension $d$ ,
sparsity $p \in [0, 1]$ , noise std. $\sigma$

**Output:** Wide features $X_{wide} \in \mathbb{R}^{n \times d}$

1: Sample weights $W \in \mathbb{R}^{m \times d}$ with $W_{ij} \sim \mathcal{N}(0, 1)$
2: Sample mask $M \in \{0, 1\}^{m \times d}$ with $M_{ij} \sim \text{Bernoulli}(p)$
3: Compute wide features $X_{wide} \leftarrow X(M \odot W)$
4: Sample noise $N \in \mathbb{R}^{m \times d}$
   with $N_{ij} \sim \mathcal{N}(0, \sigma \sigma_j)$ and $\sigma_j = \text{std}(X_{wide_{:,j}})$
5: Add noise $X_{wide} \leftarrow X_{wide} + N$
6: **return** $X_{wide}$

Figure 2: Pseudocode (left) and illustration (right) for sampling from our HDLSS prior.

Specifically, we describe our procedure in Algorithm 1, which takes as input the features $X$ of a dataset, during training sampled from the TabICL prior with a moderate feature count $m$, and then artificially *widens* it to $d \gg m$ dimensions. For this, we first sample a masked linear layer (lines 1 and 2) with sparsity $p$ and, second, apply this sparse linear projection to obtain $X_{wide}$ (line 3). Afterwards, we add Gaussian distributed noise (line 4). With this procedure, we can generate thousands of new features highly correlated to the original feature set, mimicking HDLSS data.

Our procedure can generate new features that form correlated clusters as new features depend on only a subset of the original features. The sparsity parameter $p$ controls this structure: small values yield new features influenced by few or no originals, resulting in sparse correlation patterns, whereas large values produce new features that are mixtures of many originals, leading to dense correlation patterns. Figure 3 compares real-world HDLSS biomedical data (a) with synthetic datasets (b–f), with $p = 0.02$ showing the closest match to the real correlation structure.

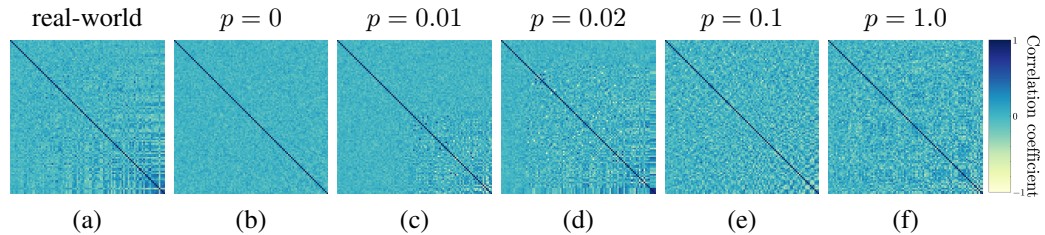

| real-world | $p = 0$ | $p = 0.01$ | $p = 0.02$ | $p = 0.1$ | $p = 1.0$ |

(a)          (b)          (c)          (d)          (e)          (f)

Figure 3: Feature correlation maps for (a) mRNA gene expression data and (b–f) synthetically generated datasets with different sparsity values $p$. We compute Pearson correlation for 100 randomly sampled features and sort them by average absolute correlation.

### 4.2 CONTINUED PRE-TRAINING

For our continued pre-training setup we start from the original TabPFNv2 classifier checkpoint[2] and updated all parameters during training. We used AdamW (Loshchilov & Hutter, 2019) (using a weight decay of $1 \times 10^{-4}$ and a learning rate of $1 \times 10^{-5}$) with linear warm-up, cosine decay, and gradient norms clipping to $1.0$. We used a batch size of 16, reducing it to 8 for training runs with over 5,000 features due to memory constraints. Training and validation were performed using cross-entropy loss. The generated datasets of the TabICL prior had up to 10 classes (to match TabPFNv2's limitations), 40 to 400 samples, and 50 to 350 features which we then widened using Algorithm 1. The number of features as parameter of Algorithm 1 was uniformly sampled between 200 and $d$ features with $d \in \{1{,}500; 5{,}000; 8{,}000\}$. With a probability of 0.5, the original features were appended to the final dataset. Sparsity and noise level were uniformly sampled with $p \in [0, 0.05]$ and $\sigma \in [0, 1]$ following our analysis visualized in Figure 3. We denote the resulting models as TabPFN-Wide-*, where * indicates the maximum number of features used during training.

For model selection, we used two real-world datasets (*COAD* and *GBM*; see description below). For each dataset, we used three different omic feature sets (only mRNA, only methylation, and concatenated mRNA + methylation + miRNA), giving rise to six HDLSS tasks. We use the model with the lowest average validation loss across these tasks. Interestingly, our continued pre-training required relatively few synthetic datasets, with the model's validation performance already approaching convergence after just 32,000 datasets while the final models continued improving until having seen up to 165,000 datasets.

### 4.3 FEATURE-WISE INTERPRETABILITY VIA ATTENTION MAPS

To gain insights into TabPFNv2's inference, we analyze attention maps, focusing on attention towards the label as a proxy for feature importance. This requires that each transformer (token) column corresponds to a dataset feature. By default, TabPFNv2 groups features, adds distribution-dependent features, or may remove features impairing a token-to-feature mapping. To address this, we disabled these modifications for training as well as our biomedical datasets and interpretability analyses.

---

[2]See Hugging Face model; Runtime complexity remains unaffected, thus, to satisfy higher resource demands for continued pre-training we used 4 NVIDIA H100 GPUs with a combined memory of 320GB.

Attention maps are an intermediate step of the original dot-product attention computation (Vaswani et al., 2017) and we refer to the matrix $\boldsymbol{A}$ in Equation (1) as "attention map", with query matrix $\boldsymbol{Q}$, key matrix $\boldsymbol{K}$, value matrix $\boldsymbol{V}$, and key vector dimensionality $d_{key}$:

$$\text{Attention}(\boldsymbol{Q}, \boldsymbol{K}, \boldsymbol{V}) = \text{softmax}\left(\frac{\boldsymbol{Q}\boldsymbol{K}^T}{\sqrt{d_{key}}}\right)\boldsymbol{V} = \boldsymbol{A}\boldsymbol{V}. \tag{1}$$

To interpret attention maps as an indicator of feature importance, we consider only TabPFNv2's feature-wise attention, disregarding the sample-wise attention. Since the embedded labels are appended before the forward pass, the attention value towards the label corresponds to the attention map's last row excluding the label index.

Furthermore, we average the attention maps across all samples, heads, and layers (similar to prior work by Ye et al. (2025)). We acknowledge that attention maps can vary substantially across these dimensions. However, this approach aligns with the intuition that features identified as relevant by the model across numerous samples, heads, or layers are those most indicative of importance (as we also show in our empirical results). In the following, the term "attention score" of a feature refers to its average attention to the label column.

## 5 EXPERIMENTS AND RESULTS

We now turn to an empirical evaluation. First, we study TabPFN-Wide's performance in two settings: (a) real-world HDLSS biomedical datasets (Section 5.1) and (b) standard benchmark tasks for predictive tabular machine learning (Section 5.2). Then, we assess its interpretability in Section 5.3.

**Datasets and Evaluation Protocol.** We use machine learning–ready TCGA datasets differing from raw TCGA data by already being normalized, quality-checked, and otherwise pre-processed. We use six datasets (two of which for model selection): *COAD*, *LGG*, and *OV* published by Yang et al. (2025) and *BRCA*, *SARC* and *GBM* by Rappoport & Shamir (2018). Table A1 provides details of the corresponding table structures. Using early integration, we concatenate all omic types (mRNA, methylation, CNV (if present), and miRNA) along the feature axis, yielding datasets with up to 60,000 features. In addition to these real-world datasets, we also evaluate on 21 benchmark tasks (with $\leq 10,000$ samples and $\leq 500$ features) introduced by *TabArena* (Erickson et al., 2020).

Unless stated otherwise, all models were evaluated using all features. In settings where we need to apply feature reduction, we recursively merge features based on the minimal Euclidean distance of pairs of feature vectors (as demonstrated to be appropriate in preliminary analyses, see Appendix A.2). However, we note that we aim to avoid feature reduction methods to retain feature-wise interpretability and solely explore it to compare model performance across different feature counts.

Alongside the foundation models TabPFNv2 and TabICL, we evaluate other baseline models, including the pre-tuned neural network RealMLP-TD (Holzmüller et al., 2025) as well as classical tree-based machine learning techniques like random forest and XGBoost (Chen & Guestrin, 2016). Ensembling was not used for TabPFN-Wide, TabPFNv2, TabICL, and RealMLP-TD.

We perform 5-fold cross-validation for our biomedical datasets to compute AUROC, AUPRC, and accuracy. For the TabArena datasets we follow the original evaluation protocol and compute AUROC using a 3-fold cross-validation repeated 3 or 10 times, depending on dataset size.

### 5.1 RESULTS ON REAL-WORLD WIDE DATASETS

**TabPFN-Wide shows superior performance across real-world HDLSS datasets.** We first evaluated our models on the test set of 4 TCGA cancer datasets not used for validation. The average AUROC scores in Table 1 highlight the strong capabilities of TabPFN-Wide. While tree-based methods exhibit stable performance, our model achieves superior results. TabPFNv2 and TabICL exhibit inferior performance consistent with the fact that they were not trained for such extreme feature counts. RealMLP-TD, trained on each dataset separately, yields comparable although slightly inferior AUROC results to TabPFN-Wide demonstrating that it also effectively handles HDLSS data. We provide further results using different metrics in Appendix A.5, showing that TabPFN-Wide shows strong performance on AUPRC as well, while exhibiting marginally weaker accuracy values compared to other models.

| Dataset | | LGG | OV | BRCA | SARC |
|---|---|---|---|---|---|
| #features | | 60,664 | 60,443 | 26,577 | 26,577 |
| | 1.5k | **0.989** ± 0.010 | **0.986** ± 0.006 | 0.978 ± 0.002 | **0.954** ± 0.005 |
| TabPFN-Wide | 5k | 0.987 ± 0.008 | 0.985 ± 0.006 | **0.984** ± 0.002 | 0.950 ± 0.007 |
| | 8k | **0.989** ± 0.009 | 0.983 ± 0.006 | 0.983 ± 0.000 | 0.953 ± 0.003 |
| TabPFNv2 | | 0.875 ± 0.010 | 0.899 ± 0.005 | 0.884 ± 0.004 | 0.902 ± 0.010 |
| TabICL | | 0.943 ± 0.010 | 0.718 ± 0.011 | 0.943 ± 0.004 | 0.863 ± 0.019 |
| R. Forest | | **0.989** ± 0.007 | 0.968 ± 0.003 | 0.982 ± 0.003 | 0.942 ± 0.017 |
| XGBoost | | 0.985 ± 0.008 | 0.971 ± 0.006 | 0.981 ± 0.002 | 0.929 ± 0.018 |
| RealMLP-TD | | 0.987 ± 0.009 | 0.982 ± 0.005 | 0.981 ± 0.004 | 0.952 ± 0.016 |

Table 1: Average AUROC (±SEM) scores on 4 real-world multiomics datasets (higher is better). We compare TabPFN-Wide, using up to 8k features for continued pre-training to TabPFNv2 and other baselines. We boldface the best values in each column.

To enable a systematic comparison of the models across a fixed set of feature counts, we applied feature reduction. Figure 4 shows the strong relative performance for all TabPFN-Wide variants compared to a random forest. While all models perform similar with heavily reduces feature sets, the performance of TabPFN and TabICL drastically declines while TabPFN-Wide's performance stays robust. Since it performs similarly with and without feature reduction, TabPFN-Wide appears to capture the correct signal (see Appendix A.4). Notably, TabPFN-Wide exhibits competitive performance even with feature counts far exceeding those seen during continued pre-training.

Interestingly, increasing the maximum width of synthetic datasets used during continued pre-training from 1,500 to 8,000 exerts only a minor influence on cancer subtype classification performance (first three rows in Table 1). Hence, further evaluation is needed to assess potential benefits of training on wider data, especially given the quadratic rise in complexity from increasing the number of features during training.

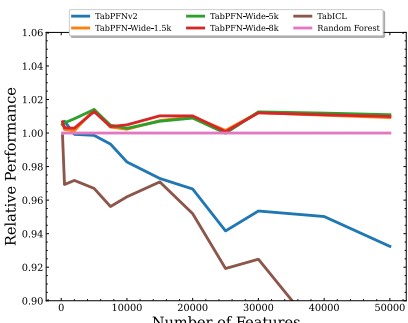

Figure 4: Average relative performance to random forest (pink) for up to 4 multiomics datasets. Higher is better. See Appendix A.4 for detailed results.

## 5.2 RESULTS ON STANDARD BENCHMARKS AND WIDENED ADAPTATIONS

**TabPFN-Wide performs on par with TabPFNv2 on TabArena Benchmark.** Figure 5a compares TabPFNv2 and TabPFN-Wide showing that our continued pre-pretraining impacts performance on standard benchmarks negligibly. This suggests that there is no indication for catastrophic forgetting.

To further explore the capabilities of our model and given the low number of HDLSS tasks present in tabular benchmarks, we generated wide datasets based on OpenML (Bischl et al., 2025) datasets. In a first step, we therefore selected datasets from the TabArena (Erickson et al., 2025) and the AutoML benchmark (Gijsbers et al., 2019) with low sample sizes ($\leq 2,500$) and $\geq 8$ numerical features. This increases the probability that multiple features are predictive in the modified dataset. To widen a dataset, we apply the same mechanism as described in Algorithm 1 on all numerical features. We explore two settings: (a) *needle-in-a-haystack* where we add noise features ($p = 0$, with the original features included) and (b) *feature smearing* where the signal is distributed across many features ($p \in 0.02, 0.25, 0.5$, without original features). Refer to Appendices A.6 and A.7 for further details of this procedure. We also include results for TabPFN-Wide-1.5k and -8k showing only minor differences to TabPFN-Wide-5k. Therefore, we limit evaluation to TabPFN-Wide-5k as a robust default setting.

**Needle in a haystack.** We compare TabPFN-Wide to baseline methods on a noise-filtering task. Precisely, we augmented the datasets with Gaussian noise features up to a total of 30,000 features,

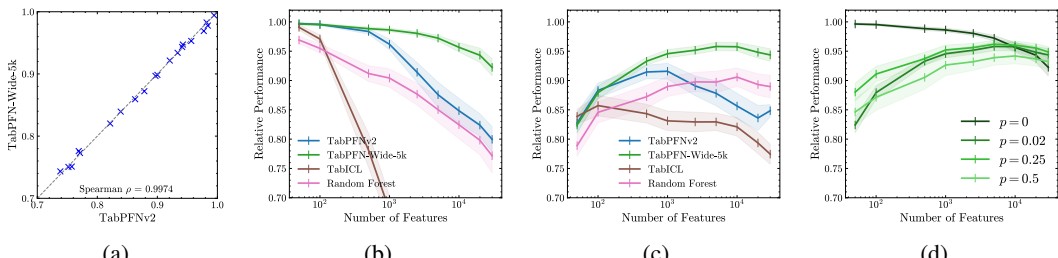

(a)            (b)            (c)            (d)

Figure 5: (a) AUROC for TabPFN-Wide-5k vs TabPFNv2 on 21 TabArena classification tasks with $\leq 10{,}000$ samples and $\leq 500$ features. (b-c) Average AUROC (relative to TabPFNv2 evaluated on the original dataset) on a set of 13 widened datasets: (b) *needle-in-a-haystack* and (c) *features-smearing* (see text for further details). (d) TabPFN-Wide-5k's performance for different sparsities. $p = 0$ corresponds to TabPFN-Wide-5k's curve in (b), and $p = 0.02$ in (c).

requiring the model to effectively isolate true signal features to achieve accurate predictions. As depicted in Figure 5b, our model (green line) is nearly unaffected by noisy features, resulting in only a slight performance decrease relative to TabPFNv2's performance on the original datasets. This highlights that TabPFN-Wide can pinpoint relevant features making up as little as $0.03\%$ of all input features, i.e., the needle in the haystack. For TabPFNv2, performance begins to decline relative to our model at around $1{,}000$ features, demonstrating its focus on datasets with $\leq 500$ features. TabICL's performance decreases even more rapidly for $> 100$ features showing the model's inability to reliably filter noise from signal.

**Feature smearing** was evaluated for different sparsities with an emphasis on $p = 0.02$ due to its realistic correlation structure. However, given the small sparsity, a substantial proportion of the features consist solely of noise (see Figure 3). For small feature counts in particular, this leads to decreased performance, as not all original features may be represented in the projected feature space. Visualized in Figure 5c, TabPFN-Wide shows the best curve of all models reaching on average about $95\%$ of TabPFNv2's performance on the original datasets. This is additionally demonstrated by Figure 5d that shows a similar curve course for sparsities up to $p = 0.5$ for our model.

### 5.3 INTERPRETABILITY

To begin our interpretability analysis, we evaluated the model on synthetically widened datasets, allowing us to assess whether attention scores reflect feature importance. Furthermore, these controlled datasets also allow us to identify, which features are expected to be predictive. We again conducted (a) *feature smearing* and (b) *needle-in-a-haystack* widening expecting our model to assign the highest scores to the original features and separate signal from noise. As described in Section 4.3, we extract the attention scores for each feature during inference and average them to obtain a single value. The generated datasets contain $2{,}000$ features and are derived from the QSAR biodegradation dataset (OpenML ID 1494). For visualization, we use correlation maps with features ordered by attention score allowing signal and noise features to be distinguished.

**Features with higher attention scores are more predictive than features with lower scores.** For the *feature smearing* dataset, Figure 6a shows that features with little correlation (upper left) can be distinguished from increasingly correlated features (lower right). Therefore, noisy features have low attention scores, while signal-rich features receive higher scores. The *needle-in-a-haystack* experiment further illustrates this: Figure 6b shows that the features with the highest attention scores correspond to those from the original dataset. Hence, the model not only successfully distinguished between noisy and predictive features to yield competitive performance (see Section 5.2), but this separation is also mirrored in the corresponding attention scores. These findings provide promising evidence that attention scores from TabPFN-Wide reflect feature importance and, consequently, represent a viable approach for interpretability. Results using TabPFNv2 (see Appendix A.8) show weaker separation of noise and signal, consistent with its lower performance on wide datasets.

Having evidence that attention maps yield useful insights in feature importance, we return to our real-world cancer datasets and validate the biological relevance of our model's attention scores by

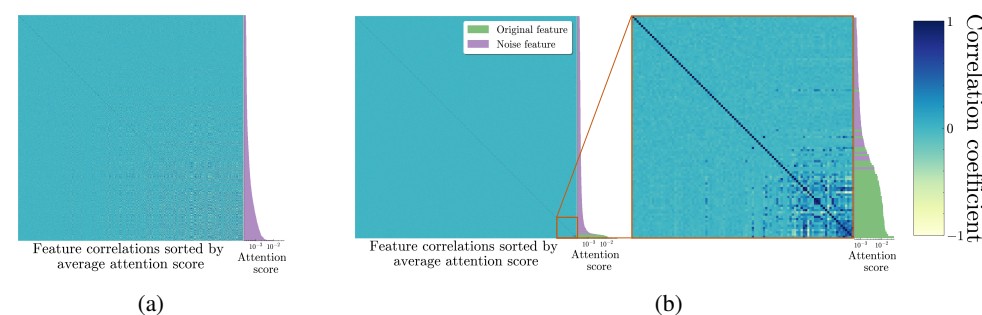

Figure 6: Correlations of 2,000 features sorted by their attention score. (a) *feature smearing* with $p = 0.02$ and $\sigma = 1$. (b) *needle-in-a-haystack*.

retrieving the features with the highest attention scores for subtype classification. Since mRNA is the most studied modality among the different omic types, we focus on the mRNA data. High correlation between genes complicates the task, since features that are presumably predictive are not necessarily causal.

**TabPFN-Wide identifies important biomarkers for different cancer subtypes.** We extracted the 10 genes with the highest attention scores from each dataset and examined their biological relevance according to literature (see Appendix A.9 for details). For breast cancer data (BRCA), all of these genes have known links to breast cancer, confirming their biological relevance and validating our method. Genes such as *FOXC1*, *ERBB2*, *PPP1R14C*, and *NDC80* are directly connected to certain subtypes of breast cancer, aligning well with the subtype classification task addressed by the model. However, in other datasets fewer features could be validated by this literature review (3/10). This may indicate that these cancer types are not as well studied as breast cancer, hinting at potentially undiscovered relationships, though variability in attention maps cannot be ruled out. Nevertheless, we believe these exciting results support the usefulness of attention maps as interpretability tools.

## 6 CONCLUSION

We introduce TabPFN-Wide, developed by continuing pre-training of TabPFNv2. To the best of our knowledge, it is the first tabular foundation model that handles HDLSS data without feature reduction and is the first application of continued pre-training to extend tabular foundation model capabilities. It achieves state-of-the-art performance on real-world and synthetic HDLSS data while simultaneously maintaining performance on small datasets. Furthermore, we show that attention scores, calculated within the transformer architecture, are indicative of feature importance and, thus, serve as an inherent interpretability method.

**Limitations.** Currently, our HDLSS prior is designed and validated only for continued pre-training of TabPFNv2. Initial attempts to train TabICL in the same manner were unsuccessful, raising the question of whether an adapted prior could solve this, or whether TabICL's architecture is inherently unable to handle HDLSS data (see Appendix A.3). Moreover, since the architecture of TabPFNv2 is unchanged, our model is limited by the (Flash-)attention mechanism's complexity and high memory requirements, restricting increases in the number of samples or features. Additionally, the attention map analysis may have limited significance. Although this approach is highly accurate for synthetic problems where the ground truth is known (i.e., needle-in-a-haystack tasks), its applicability to realistic biomedical datasets should be interpreted with caution.

**Outlook.** Since our model is currently based solely on the TabPFNv2 classifier, our approach seeks further validation from continuing pre-training of the regressor model. The prior setup is strongly inspired by the type of data faced in the biomedical domain which raises questions about whether a more diverse or sophisticated HDLSS prior allows for the creation of an even better TabPFN-Wide. While our findings suggest that attention scores are a valid approach for inherent interpretability, a systematic study of the strengths and weaknesses is pending. Overall, we show that continued pre-training has the potential to extend the capabilities of pre-trained models, like TabPFNv2, paving the way for resource-efficient generation of "patched" model versions for other dataset characteristics.

## ETHICS STATEMENT

This study makes use of data generated by The Cancer Genome Atlas (TCGA) project, which is managed by the National Cancer Institute and the National Human Genome Research Institute. All TCGA data are de-identified and publicly available in accordance with the TCGA data access policies, and therefore no additional institutional review board approval was required for this research. We complied with all relevant guidelines for the use of human genomic data and adhered to the TCGA data usage policies. No attempt was made to re-identify participants, and all analyses were performed exclusively on the provided de-identified datasets. Furthermore, only synthetic data were used for training our models. Thus, no patient information was exposed during model development, and the model weights cannot be traced back to any individual or dataset.

## REPRODUCIBILITY STATEMENT

We have made efforts to ensure the reproducibility of our work. Details of the model, training procedure, and evaluation setup are described in Sections 4 and 5 of the main paper. In addition, we include an anonymized codebase in the supplementary materials, containing our training setup and scripts, the implementation of Algorithm 1, and evaluation scripts for both the multi-omics data and the tabular benchmark datasets. We also provide data loading code with the corresponding pre-processing steps. The datasets used in our experiments are described in Appendix A.1 and are publicly available, with usage instructions given in the supplementary material. Finally, we provide the model weights for all trained models, along with code to load and apply them.

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

## LLM USAGE

Large Language Models (LLMs) were used to support the paper writing process. We used OpenAI's ChatGPT-4 and -5 to polish writing, increase conciseness of sentences, and retrieve recommendations for rewriting to increase readability and the flow of the paper. We did not use LLMs to generate any content nor did we use it for interpretation / analyses of the results. All outputs of the LLMs were thoroughly reviewed and checked before including them into the paper to guarantee that the meaning and intent stayed unaffected.

## A  APPENDIX

### A.1  DATA OVERVIEW

Table A1 gives an overview of the number of samples and features of the used datasets. Furthermore, it shows which molecular measurements are available for which dataset. Datasets provided by Yang et al. (2025) (COAD, LGG, OV) have 4 different omics: mRNA gene expression data (mRNA), copy number variation data (CNV), methylation data (Methylation) and micro RNA data (miRNA). MRNA, CNV, and methylation features are measurements corresponding to human genes. For our usage, we concatenated all different omics resulting in up to 60,000 features. Datasets provided by Rappoport & Shamir (2018) consist of less features due to missing CNV data and lower number of features for methylation data.

|  | Patients | mRNA | CNV | Methylation | miRNA | All |
|---|---|---|---|---|---|---|
| LGG (low grade glioma) | 247 | 14,260 | 21,104 | 24,979 | 321 | 60,664 |
| OV (ovarian cancer) | 284 | 14,229 | 21,104 | 24,797 | 313 | 60,443 |
| COAD (colon adenocarcinoma) | 260 | 17,261 | 19,551 | 19,052 | 375 | 56,239 |
| BRCA (breast cancer) | 440 | 20,531 | N/A | 5,000 | 1,046 | 26,577 |
| SARC (sarcoma) | 259 | 20,531 | N/A | 5,000 | 1,046 | 26,577 |
| GBM (glioblastoma) | 274 | 12,042 | N/A | 5,000 | 534 | 17,576 |

Table A1: Number of samples and features for all used datasets. Datasets used for model selection are marked in green.

### A.2  COMPARISON OF DIFFERENT FEATURE REDUCTION TECHNIQUES

In preliminary experiments, we tested the performance of TabPFNv2 on our real-world HDLSS datasets reduced with different feature reduction methods. Since this is not our main priority, we focused on simple approaches offered by *sci-kit learn* (Pedregosa et al., 2011). Although we tested both supervised (label-based) and unsupervised feature reduction methods, our preference was for the unsupervised approaches, as they better mitigate the risk of overfitting in HDLSS settings. For biomedical data, a common approach is to cluster by correlation (Langfelder & Horvath, 2008) which we compared against clustering by lowest Euclidean distance between feature vectors and reduction using the feature importance weights from fitted machine learning models. Given that Euclidean distance-based clustering frequently outperforms the correlation-based approach for our data (see Figure A1) and achieves performance comparable to supervised methods, we adopted this strategy for our analyses.

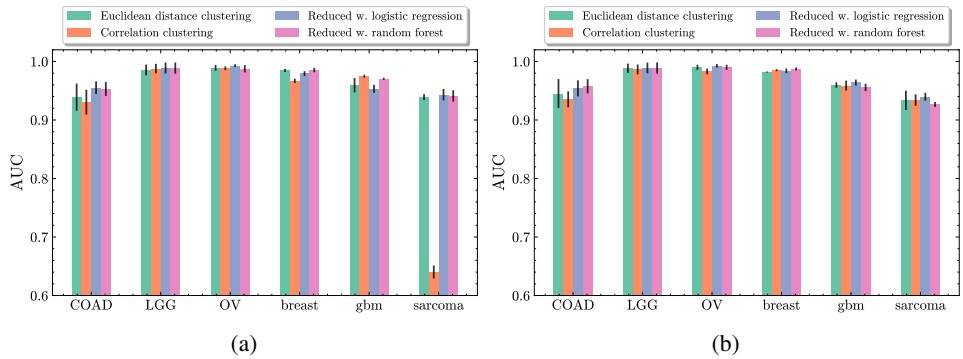

Figure A1: AUROC of TabPFNv2 evaluated on different datasets reduced to (a) 500 features and (b) 2,000 features using different techniques.

## A.3 TRAINING OF TABICL WITH HDLSS PRIOR

We tried training TabICL (Qu et al., 2025) with the same training setup as for TabPFN-Wide. However, the model's training performance did not improve, suggesting that our HDLSS prior may not be effective for TabICL. Whether this arises from TabICL's architectural setup which could make it unsuitable for HDLSS data in general or whether changes to the prior / continued pre-training could mitigate this problem, remains open for future research.

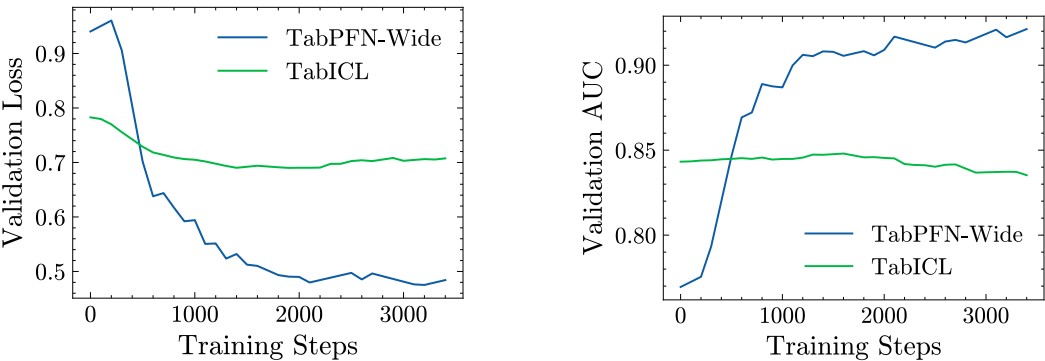

Figure A2: Development of validation loss (left) and validation AUROC (right) for TabICL vs. TabPFN-Wide when training with the same HDLSS prior.

## A.4 DETAILED RESULTS FOR ALL MULTIOMICS DATASETS

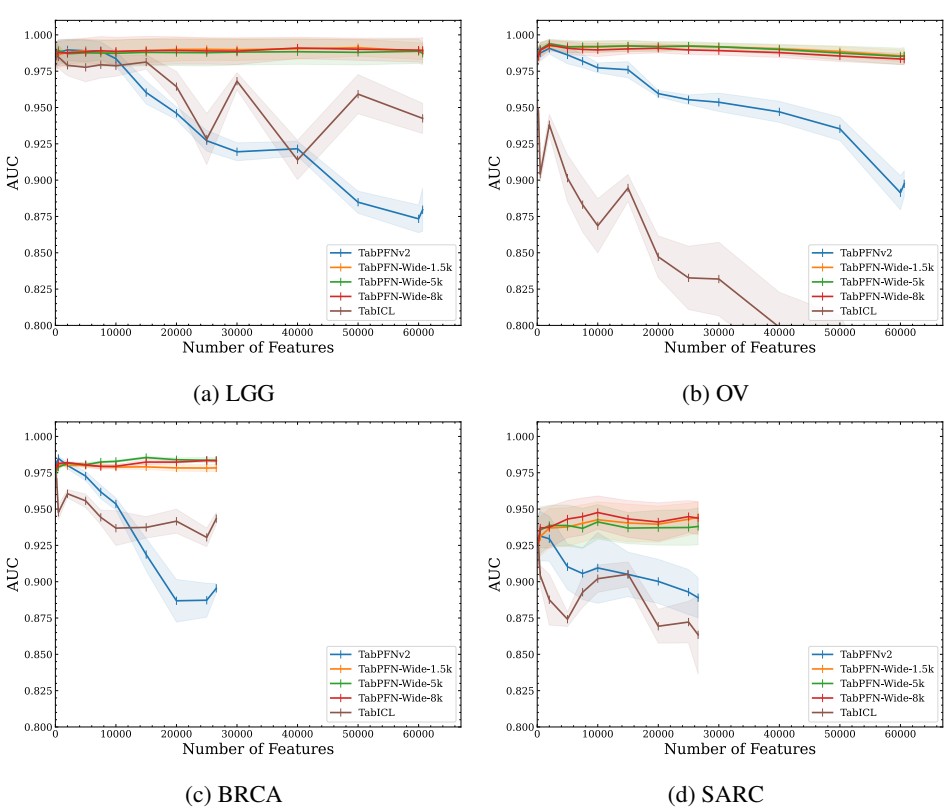

Figure A3: Single results for all datasets with feature reduction applied. The axis were chosen such that the differences in feature numbers and AUROC scores becomes comparable.

## A.5 DIFFERENT METRICS ANALYSIS

We also calculated different metrics for the evaluation on our multi-omics datasets to gain a comprehensive view and address issues posed by using AUROC only.

| Dataset | | LGG | OV | BRCA | SARC |
|---|---|---|---|---|---|
| #features | | 60,664 | 60,443 | 26,577 | 26,577 |
| | 1.5k | $0.980 \pm 0.009$ | $\mathbf{0.965} \pm 0.009$ | $0.919 \pm 0.012$ | $\mathbf{0.838} \pm 0.026$ |
| TabPFN-Wide | 5k | $0.980 \pm 0.012$ | $\mathbf{0.965} \pm 0.015$ | $0.934 \pm 0.015$ | $0.837 \pm 0.032$ |
| | 8k | $\mathbf{0.986} \pm 0.010$ | $0.960 \pm 0.009$ | $0.933 \pm 0.006$ | $0.829 \pm 0.017$ |
| TabPFNv2 | | $0.747 \pm 0.014$ | $0.795 \pm 0.008$ | $0.753 \pm 0.014$ | $0.646 \pm 0.020$ |
| TabICL | | $0.889 \pm 0.021$ | $0.507 \pm 0.006$ | $0.817 \pm 0.006$ | $0.638 \pm 0.060$ |
| R. Forest | | $0.983 \pm 0.009$ | $0.925 \pm 0.011$ | $0.926 \pm 0.016$ | $0.776 \pm 0.025$ |
| XGBoost | | $0.976 \pm 0.011$ | $0.932 \pm 0.012$ | $0.928 \pm 0.012$ | $0.790 \pm 0.043$ |
| RealMLP-TD | | $0.980 \pm 0.012$ | $0.957 \pm 0.010$ | $\mathbf{0.940} \pm 0.008$ | $0.824 \pm 0.042$ |

Table A2: Average AUPRC ($\pm$SEM) scores of 4 multiomics datasets (higher is better). We compare TabPFN-Wide, using up to 8k features for continued pre-training (second column), to TabPFNv2 and other baseline methods and boldface the best values for each column.

| Dataset | | LGG | OV | BRCA | SARC |
|---|---|---|---|---|---|
| #features | | 60,664 | 60,443 | 26,577 | 26,577 |
| | 1.5k | $0.959 \pm 0.017$ | $\mathbf{0.898} \pm 0.019$ | $0.848 \pm 0.009$ | $0.772 \pm 0.024$ |
| TabPFN-Wide | 5k | $0.972 \pm 0.005$ | $\mathbf{0.898} \pm 0.020$ | $0.884 \pm 0.009$ | $0.760 \pm 0.024$ |
| | 8k | $0.972 \pm 0.010$ | $0.887 \pm 0.009$ | $0.859 \pm 0.006$ | $0.764 \pm 0.017$ |
| TabPFNv2 | | $0.806 \pm 0.006$ | $0.679 \pm 0.008$ | $0.651 \pm 0.012$ | $0.683 \pm 0.013$ |
| TabICL | | $0.822 \pm 0.020$ | $0.472 \pm 0.014$ | $0.768 \pm 0.008$ | $0.656 \pm 0.039$ |
| R. Forest | | $0.956 \pm 0.016$ | $0.852 \pm 0.018$ | $0.845 \pm 0.009$ | $0.756 \pm 0.029$ |
| XGBoost | | $\mathbf{0.976} \pm 0.008$ | $0.824 \pm 0.014$ | $0.873 \pm 0.012$ | $0.761 \pm 0.044$ |
| RealMLP-TD | | $0.964 \pm 0.010$ | $0.884 \pm 0.016$ | $\mathbf{0.891} \pm 0.014$ | $\mathbf{0.807} \pm 0.033$ |

Table A3: Average accuracy ($\pm$SEM) scores of 4 multiomics datasets (higher is better). We compare TabPFN-Wide, using up to 8k features for continued pre-training (second column), to TabPFNv2 and other baseline methods and boldface the best values for each column.

## A.6 BENCHMARK RESULTS FOR DIFFERENT TABPFN-WIDE MODELS

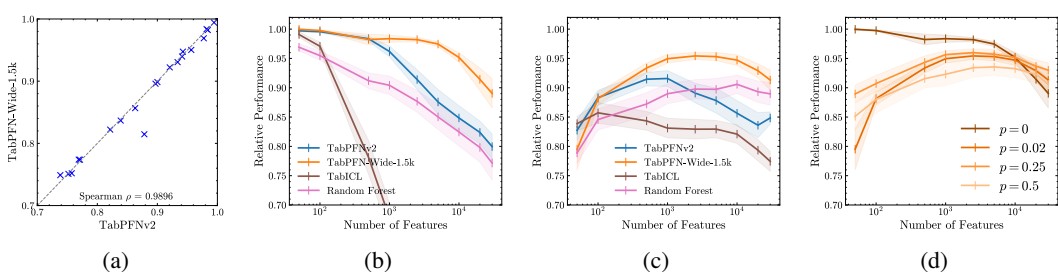

Figure A4: (a) AUROC for TabPFN-Wide-1.5k vs TabPFNv2 on 21 TabArena classification tasks with $\leq 10{,}000$ samples and $\leq 500$ features. (b-c) Average AUROC (relative to TabPFNv2 evaluated on the original dataset) on a set of 13 widened datasets: (b) *needle-in-a-haystack* and (c) *features-smearing*. (d) TabPFN-Wide-1.5k's performance for different sparsities. $p = 0$ corresponds to TabPFN-Wide-1.5k's curve in (b), and $p = 0.02$ in (c)

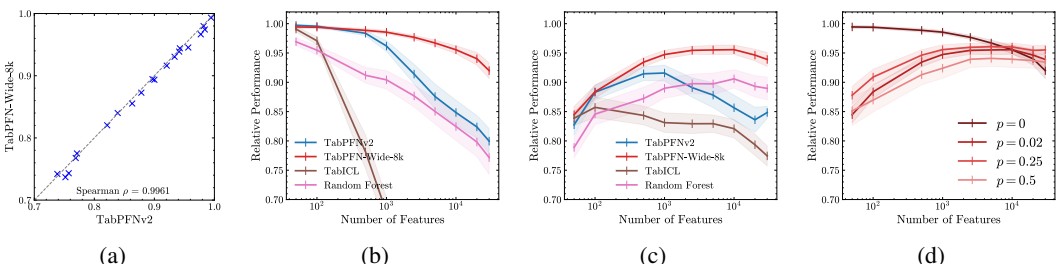

Figure A5: (a) AUROC for TabPFN-Wide-8k vs TabPFNv2 on 21 TabArena classification tasks with $\leq 10{,}000$ samples and $\leq 500$ features. (b-c) Average AUROC (relative to TabPFNv2 evaluated on the original dataset) on a set of 13 widened datasets: (b) *needle-in-a-haystack* and (c) *features-smearing*. (d) TabPFN-Wide-8k's performance for different sparsities. $p = 0$ corresponds to TabPFN-Wide-8k's curve in (b), and $p = 0.02$ in (c)

We evaluated all 3 models (TabPFN-Wide-1.5k|-5k|-8k) on the TabArena (Erickson et al., 2025) benchmark with classification datasets within TabPFNv2's sample ($\leq 10{,}000$) and feature ($\leq 500$) range. TabPFN-Wide5k has the best performance with the highest spearman correlation coefficient. TabPFN-Wide1.5k shows decent performance as well with one outlier dataset (see Figure A4). For TabPFN-Wide-8k, the performance for most datasets is slightly worse compared to TabPFNv2 showing more datasets below the diagonal compared to the other models. However, the relative and absolute performance differences are small, as seen in Figure A5. All in all, the three models maintain good performance on the TabArena benchmark, with TabPFN-Wide-5k performing best. On classification datasets within TabPFNv2's range of the AutoML benchmark (Gijsbers et al., 2024), the

results are similar with TabPFN-Wide-8k decreasing most in performance (see Figure A6). Overall, TabPFN-Wide-5k shows the highest correlation coefficient with TabPFN-Wide-1.5k's coefficient being insignificantly worse, hence overall, hinting at an inverse relationship between wider datasets during training and performance on datasets within TabPFNv2's original ranges.

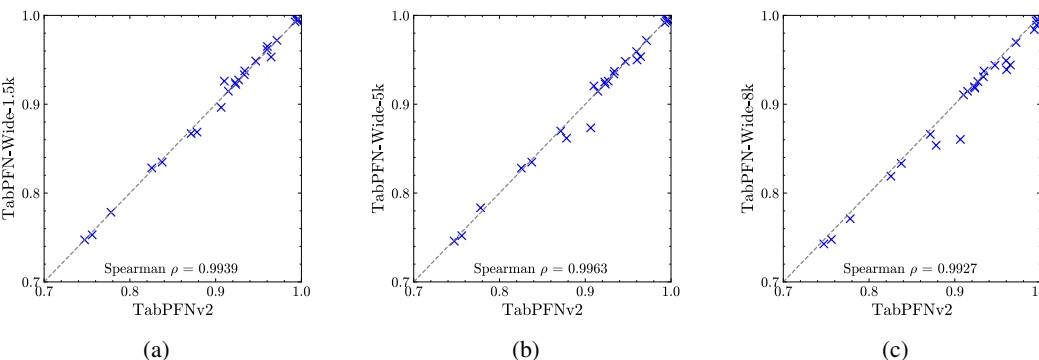

(a)  (b)  (c)

Figure A6: AUROC for TabPFN-Wide models vs TabPFNv2 on 27 AutoML benchmark classification tasks with $\leq 10{,}000$ samples and $\leq 500$ features.

For the *needle-in-a-haystack* and *feature smearing* tasks, we chose a subset of the TabArena and the AutoML benchmark. The intuition behind this selection was to evaluate TabPFN-Wide on datasets that are close to our HDLSS use case, while being synthetically generated. To include as many datasets as possible and increase the statistical significance of our analysis, we set the threshold for the maximum number of samples to $2{,}500$. Secondly, applying Algorithm 1 entails two requirements: the features must be numerical, and their number should ideally be large to ensure that the constructed features can serve as meaningful mixtures of the originals. To increase dataset inclusion, we set this threshold to at least $8$ numerical features. Since only $5$ datasets meet these requirements in TabArena, we decided to include 9 classification datasets from the AutoML benchmarks as well, resulting in a total of 13 unique datasets (1 overlapping dataset).

All models exhibit high robustness against noise for the synthetically widened datasets across different number of features and choices of the sparsity parameter $p$. This highlights the ability of TabPFN-Wide to handle diverse types of noise / features. However, while showing competitive performance on real-world HDLSS datasets (see Section 5.1) TabPFN-Wide-1.5k has a stronger performance decline compared to the other two models towards high feature counts which may stem from the reduced number of features seen during training.

## A.7  DETAILED WIDENING RESULTS FOR ALL USED DATASETS

Figures A7, A8, A9, and A10 show the results for every synthetically widened dataset that was selected for our widening experiments. The number of features refers to the absolute number of features in the dataset to allow for easier comparison regarding the width of a dataset. For Figure A7 the features of the original dataset were widened with different numbers of Gaussian noise features. For three datasets that showed missing values those were imputed to also allow for the evaluation of random forest and TabICL on them. Figures A8, A9, and A10 show the results for the datasets widened using Algorithm 1 with a sparsity of $0.02$, $0.25$, and $0.5$ respectively.

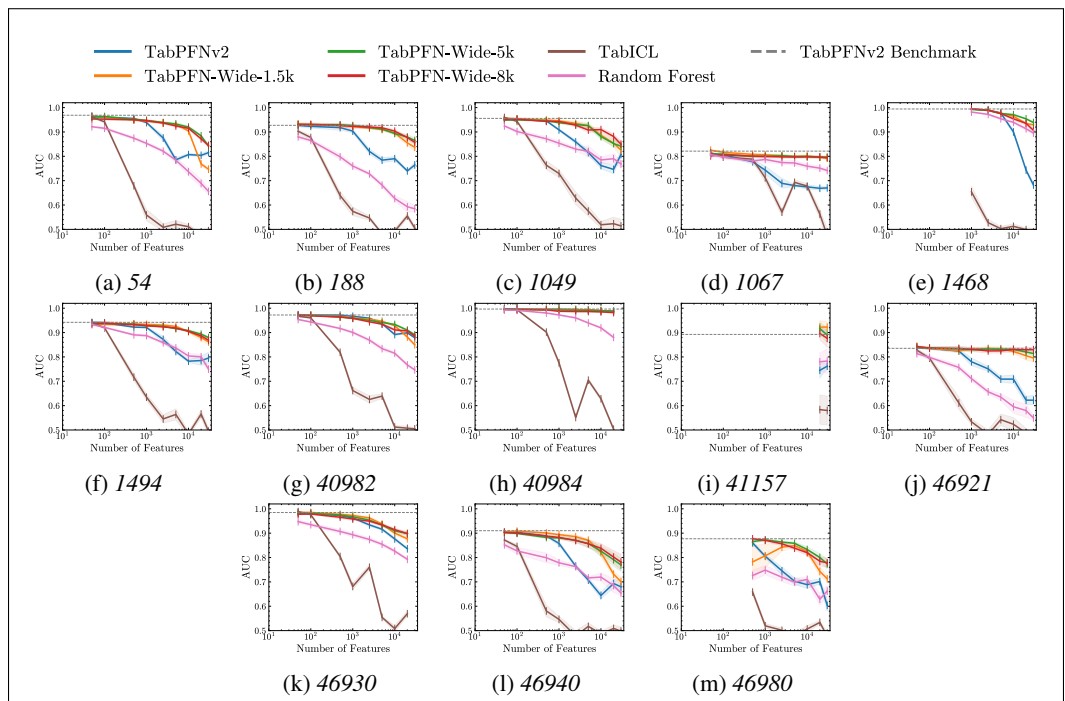

Figure A7: Detailed results for all widened OpenML datasets (*needle-in-a-haystack*). The captions indicate the corresponding OpenML dataset IDs.

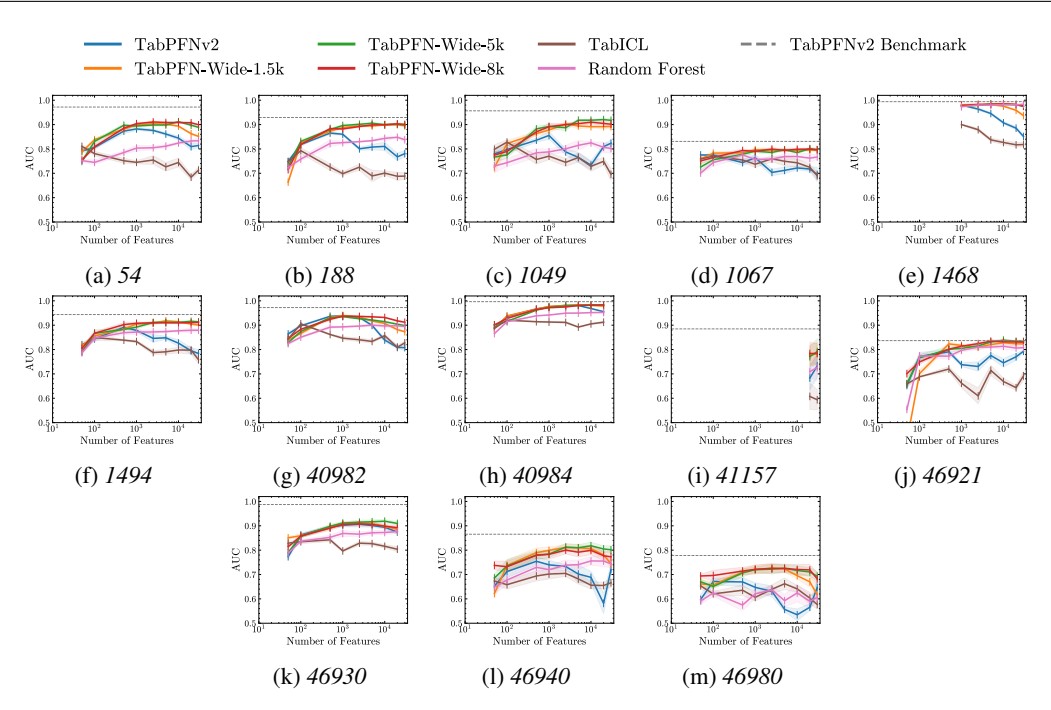

Figure A8: Detailed results for all widened OpenML datasets (*feature smearing* with $p = 0.02$). The captions indicate the corresponding OpenML dataset IDs.

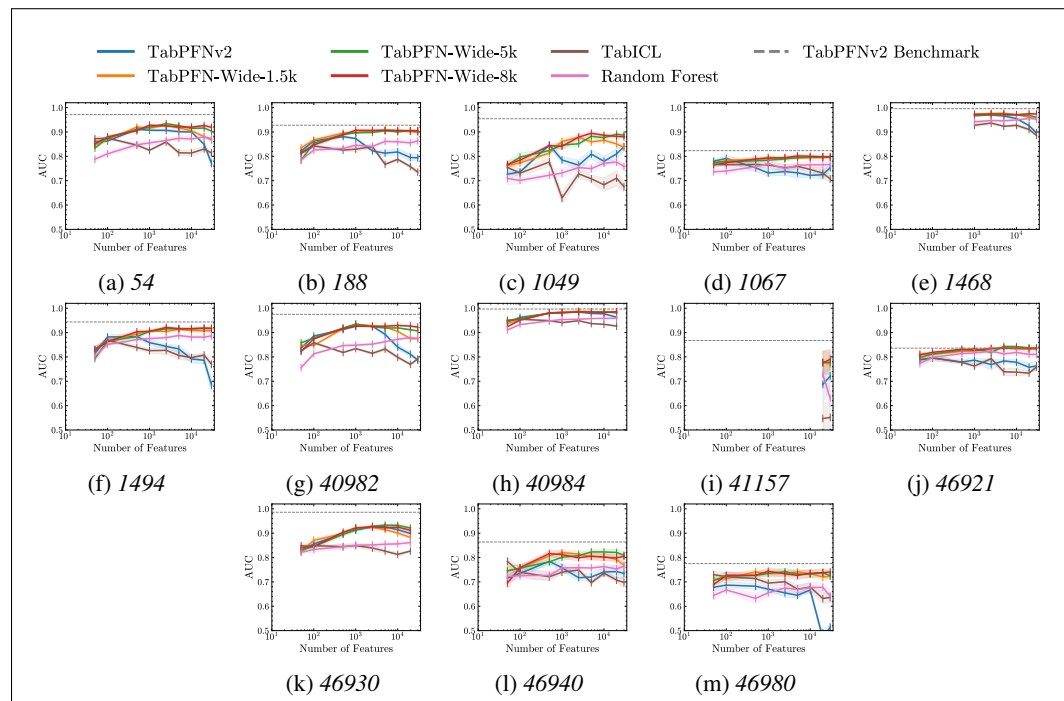

Figure A9: Detailed results for all widened OpenML datasets (*feature smearing* with $p = 0.25$). The captions indicate the corresponding OpenML dataset IDs.

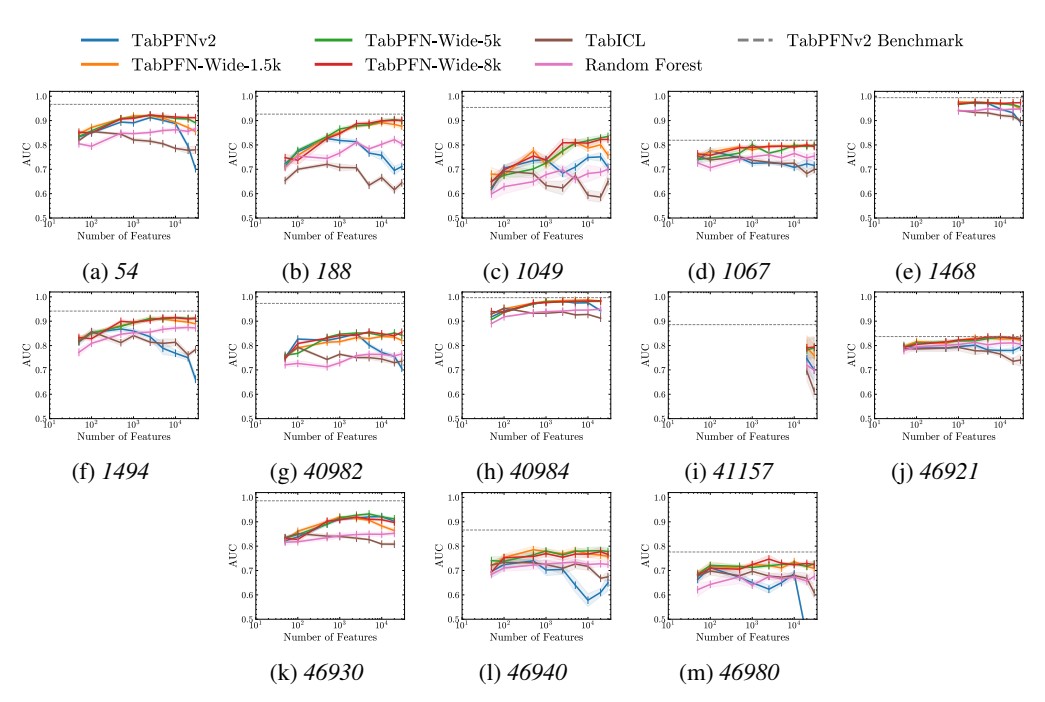

Figure A10: Detailed results for all widened OpenML datasets (*feature smearing* with $p = 0.5$). The captions indicate the corresponding OpenML dataset IDs.

## A.8 ATTENTION SCORE COMPARISON

To compare the attention scores of TabPFNv2 and TabPFN-Wide we repeated our experiments described in Section 5.3 with $10{,}000$ features with the assumption that a reduced performance coincides with a reduced interpretability of the attention scores.

Figure A11 shows the correlations of *feature smearing* datasets. TabPFN-Wide (left) shows patterns more concentrated in the lower corner whereas TabPFNv2 pattern are far more spread with even some in the upper left corner (corresponding to lowest attention scores). This indicates that our model is better at separating noise from signal for this task.

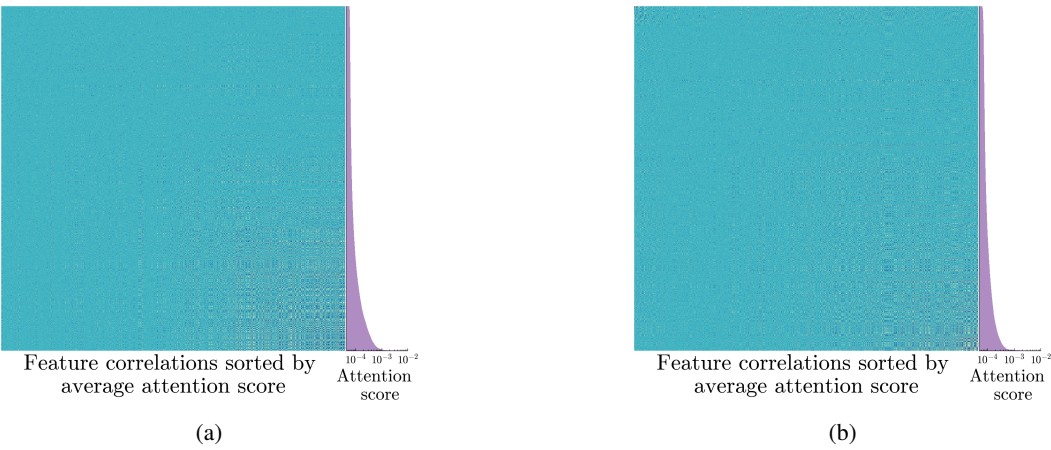

(a)                                                          (b)

Figure A11: Comparison of correlations (TabPFN-Wide (left); TabPFNv2 (right)) between features ordered by their attention score for a *feature smearing* dataset with $p = 0.02$ and $\sigma = 1$

Figure A12 shows the correlations of the 100 features with the highest attention scores for a *needle-in-a-haystack* dataset with $10{,}000$ features in total. Although TabPFNv2 is able to recover some of the original features, TabPFN-Wide identifies a larger number overall while also assigning higher average attention scores.

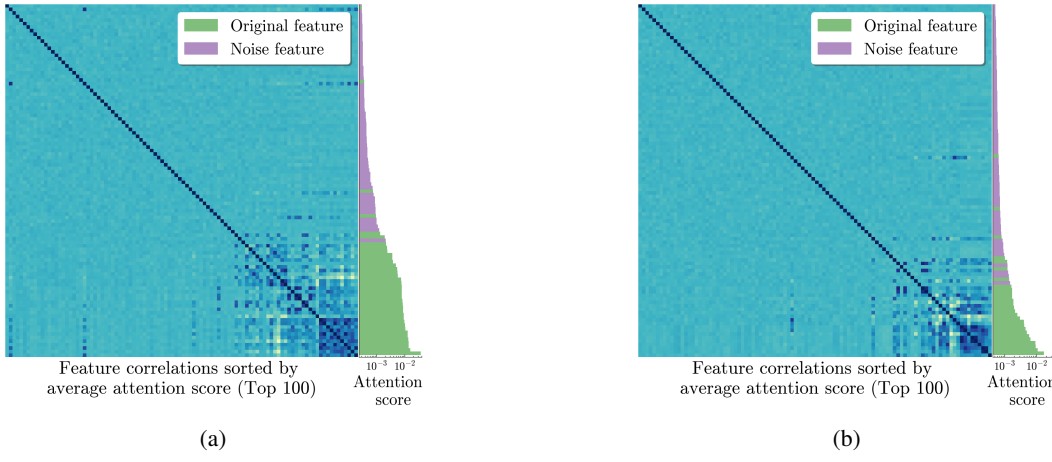

(a)                                                          (b)

Figure A12: Comparison of correlations (TabPFN-Wide (left); TabPFNv2 (right)) between the top 100 features with the highest attention scores for a *needle-in-a-haystack* dataset with $10{,}000$ features overall.

## A.9 GENES WITH HIGHEST ATTENTION SCORES

As described in Section 5.3 we analyzed the genes with the highest attention scores from our datasets with respect to literature connecting the gene with the given cancer type. We classified each gene as (i) directly associated with the specified cancer subtype, (ii) generally associated with cancer across multiple types, or (iii) having no known association with cancer. As this analysis was conducted manually, the list of citations should not be considered exhaustive. In cases where a PubMed search did not yield relevant literature, no potential associations were reported.

| Dataset | Direct Connection | General Connection to Cancer | No Known Connection |
|---------|-------------------|------------------------------|---------------------|
| LGG | RAD21 (Bady et al., 2018), MAPK4(Ren et al., 2023), NAPE-PLD(Wu et al., 2012) | | C4B, GPN1, PPP2R3C, PRKAR1B, CWF19L2, ARIH2, PORCN |
| OV | CGB7(Śliwa et al., 2019), ACSL3(Chen et al., 2016), PPA1(Li et al., 2017), CFL1(Cheng et al., 2024), CGRRF1(Lee et al., 2019), CMPK1(Zhou et al., 2017) | PHF20 (Li et al., 2013), | CFD, NAXE, PDXDC1 |
| BRCA | FOXC1 (Han et al., 2017), ERBB2 Krishnamurti & Silverman (2014), MIA (Bosserhoff et al., 1999), DSC3 (Oshiro et al., 2005), SFRP1 (Lo et al., 2006), FAM189A2 (Tsunoda et al., 2022), BLM (de Voer et al., 2015), PPP1R14C (Jian et al., 2022), NDC80 (Tang & Toda, 2015), UBE2T (Dutta et al., 2022) | | |
| SARC | TSPAN31 (Jankowski et al., 1994), MDM2(Sciot, 2021), LMOD1(Guo et al., 2015), CTDSP2(Su et al., 1997), CDK4(Su et al., 1997), METTL1(Wang et al., 2023), ADPGK(Zhang et al., 2025b), ACTG2(Lehtonen et al., 2012) | | MARCH9, FAM119B |

Table A4: Categorization of the top 10 features with the highest attention scores for datasets when performing subtype classification.

