# OpenReview forum: "TabPFN-Wide: Continued Pre-Training for Extreme Feature Counts"
_ICLR.cc/2026/Conference — Submitted to ICLR 2026_

### Official Review · Reviewer_61Dz · 2025-10-26

**Soundness:** 3
**Presentation:** 3
**Contribution:** 2
**Rating:** 4
**Confidence:** 4

**Summary:**

This mainly investigates how continued pre-training with a tailored HDLSS synthetic prior extends a tabular foundation model (TabPFNv2) to generalise to extreme feature counts (up to >50k) without sacrificing performance on standard ranges and while retaining per-feature interpretability.

**Strengths:**

1. **[Important] HDLSS synthetic prior + feature widening algorithm.** The paper proposes a masked linear projection with sparsity *p* followed by noise injection to widen moderate-dimensional synthetic datasets into *d*-dimensional HDLSS data. This produces correlated “clusters” of features that mimic real omics correlation structures.
2. **Empirical improvements on real HDLSS + benchmarks.** On 4 TCGA multi-omics datasets with 26k–60k features, TabPFN-Wide attains higher AUROC and outperforms TabPFNv2/TabICL, matching tree baselines and RealMLP-TD; on standard TabArena tasks (≤500 features) performance remains on par with TabPFNv2 (i.e., no catastrophic forgetting).
3. **Inherent interpretability via attention-to-label.** After disabling feature grouping so each token maps to a feature, average attention-to-label correlates with feature importance.

**Weaknesses:**

1. **[Important] Missing important benchmark methods for HDLSS datasets.** Although the results show that the proposed method could improve the performance of TabPFN, it would be more convincing to compare TabPFN-wide with those methods designed for HDLSS datasets. Only comparing TabPFN-wide with the existing methods may provide obscure conclusions as the other models may not achieve satisfying performance for meaningful comparisons. Specifically, I would highly suggest the authors refer to the corresponding literature for specific methods [1, 2, 3].
2. **[Important] The claimed “interpretability” could be misleading.** If I understand correctly, the “interpretability” of TabPFN-wide seems to be post-hoc analysis on the model behavior. Thus, I am unsure such post-hoc analysis can really be considered as a “characteristic” of the proposed model.
3. **Limited coverage of benchmark HDLSS datasets.** I would suggest the authors refer to [3] for more comprehensive dataset setups. In HDLSS settings, the conclusion could encode much greater variance than normal datasets, and thus it is more of a necessity to include comprehensive datasets given the paper claims TabPFN-wide to be a “foundation predictor”.
4. **Analysis on TabICL seems insufficient.** Failure to adapt TabICL may reflect prior mismatch rather than architectural limitation; I would appreciate it if the authors could expand on this.

[1] Yang, Junchen, Ofir Lindenbaum, and Yuval Kluger. "Locally sparse neural networks for tabular biomedical data." *International Conference on Machine Learning*. PMLR, 2022.

[2] Jiang, Xiangjian, et al. "ProtoGate: prototype-based neural networks with global-to-local feature selection for tabular biomedical data." *Proceedings of the 41st International Conference on Machine Learning*. 2024.

[3] Ye, Han-Jia, et al. "Revisiting Nearest Neighbor for Tabular Data: A Deep Tabular Baseline Two Decades Later." *The Thirteenth International Conference on Learning Representations*.

**Questions:**

Please refer to the "Weaknesses" section.

**Details Of Ethics Concerns:**

No ethics concerns noted on my side.

---

> ### Author Response · Authors · 2025-11-21
>
> We thank the reviewer for their thoughtful feedback and appreciate their recognition of our HDLSS synthetic-prior feature widening, our strong results on both TCGA and standard benchmarks, and the inherent interpretability offered by attention-to-label.
>
> 1) **"Missing important benchmark methods for HDLSS datasets"**
> Thank you very much for pointing this out. We will review the literature, but in the meantime please refer to our response to reviewer GAru.
> 2) **"The claimed “interpretability” could be misleading. If I understand correctly, the “interpretability” of TabPFN-wide seems to be post-hoc analysis on the model behavior."**
> We respectfully disagree that our interpretability analysis is merely a generic post-hoc characteristic. In contrast to classic post-hoc methods like Shapley or fANOVA, we obtain features importance without any additional computation. Our finding is that TabPFNv2-style models, when feature grouping is disabled, exhibit attention-to-label patterns in the attention maps that closely track feature importance, a property not previously documented for PFN architectures. While we do not claim full inherent interpretability, this behavior is a reproducible model characteristic revealed through our analysis and, to our knowledge, is the first demonstration of meaningful attention-based importance maps for TabPFN for synthetic and biomedical HDLSS tasks.
> 3) **“Limited coverage of benchmark HDLSS datasets”**
> Thanks a lot for the pointer. We would like to emphasize that we benchmark our model on tabarena, a well-established set of datasets specifically built to compare state-of-the-art tabular models. Importantly, the TabArena collection intentionally excludes outdated datasets and datasets exhibiting distributional shifts that would require fundamentally different modeling assumptions. This curation ensures a clean, robust, and widely accepted basis for comparing progress in tabular machine learning.
> 4) **“Analysis on TabICL seems insufficient”**
> Our experiments show that TabICL has a reduced performance in HDLSS regimes. As shown in Figure 4, its performance declines as more features are retained, and it also fails to recover signal in our needle-in-a-haystack tests even with fewer than 500 features (Figure 5b). A full investigation is beyond the scope of this work, but we offer two hypotheses, both speculative, as we conducted no additional TabICL-specific experiments. First, since continued pre-training failed for  TabICL (see Figure A2), it is possible that the current prior is well-suited for TabPFN-like models but not for TabICL, and that appropriate adaptations might enable TabICL to benefit from HDLSS pre-training. Second, and in our view more likely given the observed performance, TabICL’s architecture differs in a critical way: its embedding step does not incorporate labels, making it harder to separate informative from noisy features in HDLSS settings.

---

### Official Review · Reviewer_8AUV · 2025-10-28

**Soundness:** 3
**Presentation:** 4
**Contribution:** 2
**Rating:** 4
**Confidence:** 3

**Summary:**

The paper introduces TabPFN-Wide, an extension of TabPFN tailored for high-dimensional, low-sample-size (HDLSS) data. Specifically, the authors propose a continued pre-training on synthetic datasets generated from the existing SCM-based TabICL prior and widened through a sparse linear projection with added Gaussian noise. Moreover, the authors study the interpretability of TabPFN-Wide by looking at the attention maps computed within the transformer architecture.

**Strengths:**

1) TabPFN-Wide addresses an important constraint of TabPFN/TabICL-like models, meaning their inability to handle datasets with a large number of features. The addition of an interpretability analysis based on attention-to-label mechanisms provides an interesting complement to the original model.

2) The code is available and easy to use.

3) The authors provide an honest and transparent discussion of the methodology, acknowledging the limitations and potential weaknesses of their approach.

**Weaknesses:**

In my view, the contribution is partially incremental relative to the standards of ICLR:
1) The proposed continued pre-training strategy is not really a contribution of the paper, but more an experimental necessity.

2)  Interpretability analyses have been previously explored in TabPFN-like models (e.g., [1]).

3) While the proposed HDLSS prior is interesting and practically effective, its design remains largely engineering-driven and lacks a theoretical justification.

[1] Rundel, David, et al. "Interpretable machine learning for TabPFN." World Conference on Explainable Artificial Intelligence. Cham: Springer Nature Switzerland, 2024.

**Questions:**

1) Previous work has explored the fine-tuning of TabPFN-like models using synthetic data. I was wondering why the authors chose a continued pre-training strategy instead, given that the HDLSS datasets considered in this work already belong to a specific and well-defined data regime.

2) From Figure A2, it seems that the performance plateau observed for TabICL during continued pre-training might not result from an architectural limitation, but rather from the fact that the HDLSS priors generated are already well represented within the priors seen during the original meta-training.
Consequently, the lower performance on real HDLSS datasets might instead be explained by the feature reduction procedure rather than by architectural constraints. Could the authors comment on this interpretation?

3) How do the authors ensure that the HDLSS prior, which constructs $X_{\text{wide}}$ via sparse linear projections and additive noise, does not disrupt the original relationship between the original $X$ and $y$?

[1] Bühler, Magnus, Lennart Purucker, and Frank Hutter. "Towards Synthetic Data for Fine-tuning Tabular Foundation Models." 1st ICML Workshop on Foundation Models for Structured Data.

------

I find the paper interesting and technically solid, but not sufficiently novel for the standards of ICLR. Therefore, placing the work in a clear borderline area. This is why, for the moment, I propose a borderline reject.

---

> ### Author Response · Authors · 2025-11-21
>
> We thank the reviewer for their positive assessment of our work. We appreciate the recognition of TabPFN-Wide’s contribution in extending tabular foundation models to high-dimensional feature spaces, as well as the value of our attention-to-label interpretability analysis.
>
> ## Weaknesses
>
> **“The proposed continued pre-training strategy is not really a contribution of the paper, but more an experimental necessity.”**
>
> While continued pre-training is motivated by the need to handle wide-tabular data, it represents more than an experimental necessity. Our work is the first to study continued pre-training to extend TabPFN’s capability without sacrificing its original performance. We introduce a concrete and reproducible strategy for scaling to feature regimes it previously could not handle, and we show that this step yields clear empirical benefits. Moreover, the procedure provides a generalizable template that can, in principle,  be applied to other tabular foundation models facing similar distributional or dimensionality shifts.
> ### Weakness 2
>
> **“Interpretability analyses have been previously explored in TabPFN-like models”**
>
> We, agree that post-hoc methods  as discussed in the paper by Rundel et al. have been studied before. However, such methods are susceptible to manipulation (as shown by [2]). In contrast, we aim for inherent interpretability similar to GAMFormer [3]. We are the first to study TabPFN-style models from this perspective, and our results show that, despite not being designed for interpretability, TabPFN provides useful attention maps. To our knowledge, we are the first to analyze and leverage TabPFN attention maps to derive feature importances, particularly in a biomedical setting.
>
> ## Questions
>
> 1) **“I was wondering why the authors chose a continued pre-training strategy instead, given that the HDLSS datasets considered in this work already belong to a specific and well-defined data regime.”**
> Prior work on fine-tuning aims to improve performance on a given task or dataset by using this dataset to update weights [1]. Although prior work has fine-tuned TabPFN-like models on data from well-defined regimes, the HDLSS setting introduces a structural shift that TabPFN was not trained to handle. Fine-tuning on the very small real datasets is not an option due to the lack of large amounts of data. In contrast, we propose to continue pre-training on synthetic wide-tabular data. By that, the model can learn  the inductive biases needed for HDLSS problems before encountering the downstream tasks, leading to better generalization and stability.
>
> 2) **“Consequently, the lower performance [of TabICL] on real HDLSS datasets might instead be explained by the feature reduction procedure rather than by architectural constraints” **
> As noted in our response to Reviewer GAru, feature reduction was applied only to explore whether the lost signal could be recovered with fewer features (which it was not). TabICL’s performance declines as more features are retained (see Figure 4), suggesting that feature reduction even helps by bringing the data closer to the feature range it was originally trained for. Overall, it performs poorly on all features of the biomedical HDLSS datasets, indicating it is not suited for very high-dimensional data. This limitation is further highlighted by the needle-in-a-haystack experiments, where TabICL fails to separate signal from noise even with fewer than 500 features. This is likely because its embedding step does not incorporate labels, making it harder to distinguish informative from noisy features and therefore more likely hinting at architectural limitations.
>
> 3) **“How do the authors ensure that the HDLSS prior, which constructs X_wide via sparse linear projections and additive noise, does not disrupt the original relationship between the original and y?” **
> It is unclear what kind of “disruption” you mean? The widened features $X_{wide}$ are produced through linear projections of the original inputs, so they remain directly tied to the original feature space. While an individual widened feature may not fully reflect its source feature due to the added noise, the use of many such features ensures that the original signal is largely preserved (as shown in our feature smearing experiments). Consequently, this transformation maintains the underlying input–label relationship and does not affect $y$.
>
> [1]  Garg, A., Ali, M., Hollmann, N., Purucker, L., Müller, S., & Hutter, F. (2025). Real-TabPFN: Improving Tabular Foundation Models via Continued Pre-training With Real-World Data.
>
> [2] Bordt, S., Finck, M., Raidl, E., & von Luxburg, U. (2022). Post-Hoc Explanations Fail to Achieve their Purpose in Adversarial Contexts.
>
> [3] Mueller, A., Siems, J., Nori, H., Salinas, D., Zela, A., Caruana, R., & Hutter, F. (2024). GAMformer: In-Context Learning for Generalized Additive Models.

---

### Official Review · Reviewer_GAru · 2025-10-31

**Soundness:** 2
**Presentation:** 2
**Contribution:** 2
**Rating:** 2
**Confidence:** 4

**Summary:**

The paper presents TabPFN-Wide, an extension of TabPFNv2 architecture, focusing on modeling high-dimensional, but low sample size (HDLSS) tasks. The main contribution of the method is the feature-widening prior, which leverages a tree-based structural prior (similar to TabICL) but applies a sparse linear projection to the input features, thereby widening them to the desired target (and much larger) dimension. This procedure generates thousands of new features that are highly correlated with the original feature set. TabPFN-Wide uses continued pre-training, starting from a TabPFNv2 and continuing. to train the model with the new wide priors. Experiments on 4 HDLSS datasets derived from TCGA show that TabPFN-Wide performs slightly better than baselines on cancer subtyping (?) tasks as well as on standard tabular data benchmarks from TabArena. The authors also provide a study on feature interpretability derived from TabPFN-Wide.

**Strengths:**

- Addressing the HDLSS problem in biomedical applications (e.g., cancer genomics) is well-motivated and of practical importance.
- The manuscript is clear, well-written, and generally easy to follow.

**Weaknesses:**

- W1: The paper addresses an important practical challenge in tabular data learning that relates to high-dimensional, but low sample size (HDLSS) tasks. However, there is neither a discussion nor a comparison of many approaches that address these tasks. To name a few [1-7].

- W2: The experiments (and results) on the HDLSS dataset seem overly optimistic. Given that the authors have a ~60K/ ~25K dimensional problem (LGG,OV/BRCA,SARC) with ~200 samples and are still able to achieve very high performance (even across baselines). This suggests that the benchmark is oversaturated ie. that the tasks are very trivial. Even the SD, which I assume is from the 5-fold CV, seems very small for such problems. Even more, it is unclear what the actual subtyping tasks are, as they are never documented. A better benchmark would be [8], which also discusses the challenges of these tasks. Many of [1-7] have also been assessed on these data. Tabpfn-wide should also be evaluated against traditional lasso/elastic-net. They typically perform very well in such scenarios, and also allow explainable post-hoc analysis.

- W3: The correlation assumption (Clarke 2008), while valid for certain domains, doesn’t necessarily generalize to other domains/tasks [8] (often there is low correlation between estimated and true error in HDLSS scenarios). Therefore, this might not be beneficial if Tabpfn-wide is applied to tasks other than omics. Moreover, this might also be the reason why there is no significant difference in performance between the size variants of TabPfn-Wide, as well as the lack of effect on TabICL.

---
[1] Romero et al "Diet networks: Thin parameters for fat genomics” 2017

[2] Singh et al “Feature selection network on high-dimensional biological data” 2023

[3] Ruiz et al. "Tabular deep learning when d >> n by using an auxiliary knowledge graph” 2023

[4] Margeloiu et al “Weight predictor network with feature selection for small sample tabular biomedical data” 2023

[5] Bhalm et al "Concrete Autoencoders: Differentiable Feature Selection and Reconstruction” 2019

[6] Liu et al “Deep neural networks for high dimension, low sample size data” 2017

[7] Feng et al “Sparse-Input Neural Networks for High-dimensional Nonparametric Regression and Classification” 2019

[8] Kuncheva et al  “Feature Selection from High-Dimensional Data with Very Low Sample Size: A Cautionary Tale ”2020

**Questions:**

- See weaknesses
- What are the classification tasks being solved in 5.1? What's the evaluation setup (eg. train/val/test, strata etc.)? How were the baselines tuned for these tasks?
- I would appreciate some clarification on how TabPFNv2 and TabICL were run for the experiments in Section 5.1? It is stated that “Unless stated otherwise, all models were evaluated using all features” (L300). Was feature reduction used as discussed in A2, and if so, which one is the one reported in Table 1?
- The authors should consider providing the full results of the TabArena benchmark, as the figures, while helpful, are insufficient for analyzing the methods' behavior.
- How substantial is the additional computational overhead of cont. pre-training?
- It would be interesting to see how the important features identified by Tabpfn-wide compare to the ones identified by XGB and RF as they also have similar performance.

---

> ### Author Response · Authors · 2025-11-21
>
> We thank the reviewer for the detailed review and the raised questions. We are glad to hear that the reviewer shares our opinion on the high relevance of the solved biomedical tasks.
> ### Weakness 1
>
> “[...] However, there is neither a discussion nor a comparison of many approaches that address these tasks”
>
> Thank you for raising this concern. We focus on HDLSS data in the omics domain and standard, well curated benchmarks from the tabular machine learning community and will make this clear in the discussion. We fully agree that results on a similarly well curated benchmark for HDLSS data would be insightful and will investigate the provided resources.
>
> ### Weakness 2
> “The experiments (and results) on the HDLSS dataset seem overly optimistic. [...]””
>
> We agree that one could have chosen many different real world data sets. We chose the omics example, since this type of data gives much information about the disease state of a patient, is getting cheaper and cheaper and therefore will be available for many other diseases in the future, enabling precision medicine approaches. Nevertheless, the best curated data sets for omics data can usually be found in the cancer domain and to get to high enough sample sizes in different cancers the cancer subtyping problem is often the first choice. Nevertheless, even though the prediction problem is comparatively easy, we see a decline in performance for the standard TabPFN v2. On additional data sets we looked at after the submission with more challenging prediction problems we see a similar trend with TabPFN-Wide being as good as or even better than traditional baselines and TabPFN v2 being worse.
>
> ### Weakness 3
> “The correlation assumption (Clarke 2008), while valid for certain domains, doesn’t necessarily generalize to other domains/tasks [8] (often there is low correlation between estimated and true error in HDLSS scenarios). Therefore, this might not be beneficial if Tabpfn-wide is applied to tasks other than omics. [...]”
>
> Thank you for pointing this out. We will clarify this in our limitations section. Note however that we also show that the method outperforms the other methods in the needle-in-a-haystack setting (see Fig. 5 b), where we include additional noise features that are not correlated at all to any signal features showing that using TabPFN-Wide for tasks other than omics could be beneficial.

---

> ### Author Response · Authors · 2025-11-21
>
> ## Questions
> - “What are the classification tasks being solved in 5.1? What's the evaluation setup (eg. train/val/test, strata etc.)? How were the baselines tuned for these tasks?”
> The classification tasks we address are subtype-classification tasks on the four datasets not used for model selection (LGG, OV, BRCA, SARC). Subtype classification in general leverages the fact that each cancer subtype exhibits well-defined molecular and histological subtypes that reflect distinct biological origins and clinical behavior. Identifying these subtypes is clinically important, as it directly informs diagnosis and therapeutic decision-making. While Section A1 provides a brief overview of the datasets and their provenance, we can include a more comprehensive description of the datasets and all preprocessing steps in the appendix of a revised version if the reviewer finds this helpful.
> As described in Section 4.3, we employ 5-fold stratified cross-validation, which corresponds to an 80/20 train–test split in each fold. For the foundational models, a separate validation set is unnecessary because no hyperparameter tuning is performed. Similarly, we did not conduct hyperparameter tuning for XGBoost, Random Forest, or the RealMLP, but used the recommended default hyperparameter setting..
>
> - “I would appreciate some clarification on how TabPFNv2 and TabICL were run for the experiments in Section 5.1? It is stated that “Unless stated otherwise, all models were evaluated using all features” (L300). Was feature reduction used as discussed in A2, and if so, which one is the one reported in Table 1?”
> No, we did not apply feature reduction for the results on the omics datasets in Table 1. Feature reduction was used only for experiments shown in Figures 1 and 4, where we demonstrate that TabPFN-Wide maintains stable performance across different feature counts on the biomedical datasets (described in A2). We highlight that such feature reduction, as discussed in the reference [7], can substantially impact performance and interpretability. Accordingly, these results serve an analytical purpose only and should not be viewed as practically applicable.
>
>
> - “The authors should consider providing the full results of the TabArena benchmark, as the figures, while helpful, are insufficient for analyzing the methods' behavior.”
> We are happy to provide results per dataset in a revised version of the paper.
>
> - “How substantial is the additional computational overhead of cont. pre-training?”
>    Continued pre-training ran for \~10K steps on 4×H100 GPUs (\~20 h). Overall, our model saw only 160K additional dataset, which is around 0.12% of the 130,000K datasets used for initial pre-training.
>
> - “It would be interesting to see how the important features identified by Tabpfn-wide compare to the ones identified by XGB and RF, as they also have similar performance.”
> Our main focus was to extend the capability of models like TabPFN to HDLSS data. While a comprehensive comparison across interpretability methods and feature-importance metrics would indeed be valuable, particularly for studying the Rashomon effect in this setting, we believe such an analysis is substantial enough to be its own paper.

---

### Official Review · Reviewer_1pix · 2025-11-01

**Soundness:** 2
**Presentation:** 2
**Contribution:** 2
**Rating:** 2
**Confidence:** 3

**Summary:**

This paper addresses the problem of learning from high-dimensional, low-sample-size (HDLSS) data, with applications in biomedicine. The authors propose a continued pre-training strategy to adapt the pre-trained foundation model TabPFNv2 for learning from HDLSS datasets.

Specifically, the paper introduces a widening procedure to generate synthetic HDLSS data, which is then used to continue pre-training TabPFNv2, resulting in an adapted model called TabPFN-Wide. The authors also employ attention maps to analyze feature importance and enhance model interpretability. Numerical experiments demonstrate the performance of TabPFN-Wide on several datasets.

**Strengths:**

1. The paper proposes a novel continued pre-training approach to adapt TabPFNv2 for HDLSS data.

2. Empirical results indicate improved performance over baseline methods and prior foundation models.

**Weaknesses:**

1. The contribution appears incremental, and the proposed method is demonstrated only on a single foundation model (TabPFNv2).

2. The approach lacks theoretical rigor and detailed justification for several design choices.

3. The evaluation of feature importance using attention matrices is unclear and insufficiently described.

**Questions:**

The paper presents a potentially interesting idea of adapting pre-trained foundation models for domain-specific tasks involving HDLSS data. However, the current version of the paper has several shortcomings that limit its impact and clarity. My detailed comments are as follows:

1. Incremental Contribution and Limited Scope

The proposed approach appears incremental and seems to work only for TabPFNv2. Similar ideas—such as generating synthetic data or using linear neural networks with priors to augment features—have been explored previously.

Moreover, the use of averaged attention maps across samples, heads, and layers for feature interpretability has been considered in earlier studies.

It is also unclear whether the proposed feature widening via random synthetic data and continued pre-training can generalize beyond TabPFNv2 to other foundation models such as TabICL. Without such demonstrations, the generality of the approach remains uncertain.

2. Lack of Rigor and Justification

Several aspects of the proposed approach appear ad hoc or insufficiently explained. For example:

i. What motivates the use of random samples drawn from directed acyclic graphs (DAGs)?

ii. What is the role of the mask in Algorithm 1?

iii. Why is Gaussian noise added to the linear layer outputs?

iv. How does Algorithm 1 ensure that the generated synthetic data accurately mimic real-world HDLSS characteristics?

v. The selection of parameters such as the number of features (d) and sparsity (p) also lacks justification.

vi. Additionally, key details about the model architectures (TabPFNv2, TabICL) are missing. It is also unclear what constitutes the input feature matrix X in Algorithm 1—are these features derived from real datasets or synthetically generated?

3. Unclear Use of Attention for Feature Importance

The explanation of feature importance via attention matrices is inconsistent and underdeveloped.

In Section 2, authors say "the role and interpretability of attention maps are controversial in the literature" ; "attention maps may provide a coarse indication of a model’s reasoning process, they are often noisy and can erroneously emphasize irrelevant tokens", and "For TabPFNv2’s attention specifically, earlier research shows that it evolves across layers, shifting from label-focused attention in the first layers to semantically relevant attribute attention in deeper layers". Yet, in section 4.3, attention matrix is used for feature importance, while also claiming "We acknowledge that attention maps can vary substantially across these dimensions."

This contradiction weakens the interpretability claims. A more rigorous justification would strengthen this section.

---

> ### Author Response · Authors · 2025-11-21
>
> Thanks a lot for the detailed review and the raised questions. We particularly appreciate the acknowledgment of the novelty and strong empirical performance of our approach. We hope to address your concerns below and look forward to the discussion.
>
> ### 1. Incremental Contribution and Limited Scope
>
> [“only for TabPFNv2” / “generality remains unclear”] We appreciate the reviewer’s concern. Our goal is not to claim continued pre-training for solely TabPFNv2; in fact, we explicitly note in the limitations that other TFMs may benefit as well. We focus on TabPFNv2 for two practical and methodological reasons:
> (1) it is currently the most widely used state-of-the-art TFM for predictive tabular tasks (based on citations and stars on GitHub), and therefore the most relevant target for studying HDLSS behavior;
> (2) it is the only competitive architecture that preserves a 1-to-1 feature–token mapping, which is essential for analyzing feature importance and for diagnosing how feature widening affects internal representations.
> Other architectures, such as TabICL, do not maintain this relationship (and we show in Appendix A.3 that our approach does not work out of the box for TabICL; see also discussion with reviewer 8AUV).
>
> [“Similar ideas [...] have been explored previously”] We thank the reviewer for raising this point. Existing works indeed explore fine-tuning TFMs on real datasets to improve performance on a particular benchmark or downstream task [1,2]. Our approach differs fundamentally in both objective and mechanism. Rather than fine-tuning, we continue pre-training on fully synthetic, structurally controlled datasets, explicitly designed to improve TFMs' generalization to HDLSS data. This eliminates the need for, in our case, scarce real-world HDLSS datasets and enables us to target distributional properties that standard fine-tuning cannot address.
>
> ### 2. Lack of Rigor and Justification
>
> - **“i. What motivates the use of random samples drawn from directed acyclic graphs (DAGs)?”**
> DAG-based synthetic data generation is not an arbitrary design choice; it is the standard and empirically validated procedure used to pre-train TabPFNv2 and other PFN-style TFMs [3,4]. Our motivation is to follow the same generative assumptions used during TabPFNv2’s original pre-training, keeping the continued pre-training distributionally consistent. Thus, we closely follow the original pre-training pipeline to ensure that the widened datasets are otherwise similar to the datasets seen during the original pre-training.
> Importantly, our method in Algorithm 1 is agnostic to the specific data generator. The feature-widening procedure does not rely on DAG structure itself; DAGs are simply used because they are the canonical generator for TabPFNv2’s pre-training. Alternative generative processes could be incorporated with no change to the algorithm. We will clarify this in the revision.
>
>
> - **“ii. What is the role of the mask in Algorithm 1?”**
> The binary mask indicates which features are correlated with each other. Without the mask (or if all entries are $1$), each widened feature would be a mixture of all input features, producing unrealistically high global correlations. Omic and other HDLSS data exhibit sparse correlations with each feature being correlated with only a small subset of all features.  For example, features of omic data naturally form correlation clusters, being associated with a few other features and uncorrelated to the majority of features. The mask enforces this by restricting each widened feature to depend on average only on $p$% of the input features. For small values of $p$, this leads to the desired pattern of each feature being correlated to only a few other features.
>
>
> - **“iii. Why is Gaussian noise added to the linear layer outputs?”**
> We add Gaussian noise to match TabPFNv2’s original pre-training procedure, where noise is injected at each DAG edge to introduce uncertainty [5]. This is done to increase the robustness of the model against noise by diluting the relationships from features to the target and, therefore, forcing the model to adapt to the noisy environment.
>
> - **“iv. How does Algorithm 1 ensure that the generated synthetic data accurately mimic real-world HDLSS characteristics?”**
> Since the true generative process of real-world HDLSS data is unknown, no method can strictly ensure that the generated data is accurate. Nevertheless, Figure 4 shows empirically that the widened synthetic datasets reproduce correlation structures similar to our real HDLSS benchmark.

---

> ### Author Response · Authors · 2025-11-21
>
> - **“v. The selection of parameters such as the number of features (d) and sparsity (p) also lacks justification.”**
> We selected the sparsity parameter based on its ability to reproduce correlation structures observed in real-world HDLSS data (see Figure 4). While we acknowledge that this choice is not theoretically derived, we emphasize that it does not limit the validity of our results. In particular, we demonstrate that our model maintains strong performance even for settings with choices of the sparsity parameter $p$ up to $0.5$), despite being trained only on data with $p \leq 0.05$. This robustness substantially mitigates any concern regarding the exact sparsity used during training and indicates that the model generalizes well beyond the regime in which it was trained.
> Our choice of different values for the parameter d was motivated by the question of whether models trained on wider synthetic datasets would exhibit improved performance on our wide real-world datasets. Since TabPFNv2 is known to perform well only up to approximately 500 features, simply selecting d within that range would not meaningfully test this hypothesis. We therefore extended pre-training to substantially wider settings, starting at 1.5k features (a 3× increase) and going up to 8k features (a 16× increase). The upper limit of 8k was determined by available compute and the quadratic complexity of attention mechanisms. Importantly, our empirical results indicate no meaningful performance differences across these models.
>
> - **“vi. a) Additionally, key details about the model architectures (TabPFNv2, TabICL) are missing.”**
> Both TabPFNv2 and TabICL are publicly available models, and we rely on their standard implementations. To help us address this properly, could you clarify which specific architectural details you find missing?
>
> - **“vi. b) It is also unclear what constitutes the input feature matrix X in Algorithm 1—are these features derived from real datasets or synthetically generated?”**
> X refers to a synthetic dataset generated from a (in our case, DAG-based) prior used to pre-train the TFM.
>
>
> ### 3. Unclear Use of Attention for Feature Importance
> We thank the reviewer for pointing out the potential confusion. Our intention is not to claim that attention is a universally reliable interpretability mechanism. In Section 2, we discuss limitations of generally using attention for explainability to provide context and to acknowledge prior work. The statements about variability in Section 2 highlight that attention can be noisy on a per-head or per-layer basis, which is why we do not rely on individual attention maps ([6], [7]). Experiments in Section 5.3 validate this diagnostic tool in two settings (needle-in-a-haystack and feature-smearing) to highlight its practical usefulness.
>
> Overall, our use of attention is intentionally conservative and aligned with the caveats we previously articulated. Finally, and more importantly, our results and conclusions do not depend on attention being a perfect explanation mechanism. Our use of this tool serves two purposes: (1) as qualitative evidence that our continued-pretraining approach reliably identifies relevant dimensions in high-dimensional data (see also Figure A12) and (2) as a promising tool to generate hypotheses about the important features in real-world applications.
> We will clarify this in a revised version to avoid implying stronger interpretability claims than intended.
>
>
> [1] Garg, A., Ali, M., Hollmann, N., Purucker, L., Müller, S., & Hutter, F. (2025). Real-TabPFN: Improving Tabular Foundation Models via Continued Pre-training With Real-World Data.
>
> [2] Bühler M., Purucker L., Hutter F.. (2025) Towards Synthetic Data for Fine-tuning Tabular Foundation Models
>
> [3] Hollmann, N., Müller, S., Eggensperger, K., & Hutter, F. (2023). TabPFN: A Transformer That Solves Small Tabular Classification Problems in a Second.
>
> [4] Qu, J., Holzmüller, D., Varoquaux, G., & Morvan, M. L. (2025). TabICL: A Tabular Foundation Model for In-Context Learning on Large Data.
>
> [5] Hollmann, N., Müller, S., Purucker, L. et al. (2025) Accurate predictions on small data with a tabular foundation model.
>
> [6] Ye, H.-J., Liu, S.-Y., & Chao, W.-L. (2025). A Closer Look at TabPFN v2: Understanding Its Strengths and Extending Its Capabilities.
>
> [7] Rubachev, I., Kotelnikov, A., Kartashev, N., & Babenko, A. (2025). On Finetuning Tabular Foundation Models.

---

### Meta-Review · Area_Chair_ncnd · 2026-01-07

**Summary:**

This paper proposes TabPFN-Wide, extending TabPFNv2 to extreme feature counts (up to >50k) via continued pretraining with a tailored synthetic HDLSS prior, aiming to retain strong performance on standard tabular ranges while enabling wide biomedical use cases.

Across reviewers, the direction is seen as interesting and technically competent, and the empirical results on wide/biomedical settings are promising. However, the consensus is that the current submission falls short of ICLR standards primarily due to limited novelty framing, insufficient benchmarking against HDLSS-specific methods, and uncertainty around the interpretability claims.

**Reviewer Concerns:**

- Novelty and contribution clarity: Two reviewers view the contribution as incremental (continued pretraining plus widening priors) and not sufficiently novel/insightful as currently positioned.

- Benchmark completeness for HDLSS: Multiple reviewers request comparisons against representative HDLSS/biomedical tabular methods, noting that comparisons largely against general tabular baselines may not substantiate the “foundation predictor for HDLSS” claim.

- Interpretability claim: Reviewers question whether the presented interpretability is a model characteristic versus post-hoc analysis, and ask for clearer positioning and stronger evidence.

- Method justification and assumptions: Reviewers ask for stronger justification of the widening procedure, sparsity/noise choices, and clearer discussion of what the synthetic prior is (and is not) preserving.

**Reviewer Scores:**

Reviewer 1pix: 2 (reject). Likely unchanged without additional evidence beyond clarifications.

Reviewer GAru: 2 (reject). Likely unchanged without broader HDLSS benchmarking and stronger positioning.

Reviewer 61Dz: 4 (borderline reject). Could move upward with concrete added HDLSS baselines/datasets and clearer interpretability framing, but unlikely to flip based on rebuttal alone.

Reviewer 8AUV: 4 (borderline reject). Similar: could move with strengthened experiments/positioning, but current record supports borderline reject.

---

### Decision · Program_Chairs · 2026-01-26

Reject